# On the Power of the Weisfeiler-Leman Test for Graph Motif Parameters

**Matthias Lanzinger**
TU Wien & University of Oxford
matthias.lanzinger@tuwien.ac.at

**Pablo Barceló**
PUC Chile & IMFD & CENIA
pbarcelo@uc.cl

## Abstract

Seminal research in the field of graph neural networks (GNNs) has revealed a direct correspondence between the expressive capabilities of GNNs and the $k$-dimensional Weisfeiler-Leman ($k$WL) test, a widely-recognized method for verifying graph isomorphism. This connection has reignited interest in comprehending the specific graph properties effectively distinguishable by the $k$WL test. A central focus of research in this field revolves around determining the least dimensionality $k$, for which $k$WL can discern graphs with different number of occurrences of a pattern graph $P$. We refer to such a least $k$ as the WL-dimension of this pattern counting problem. This inquiry traditionally delves into two distinct counting problems related to patterns: subgraph counting and induced subgraph counting. Intriguingly, despite their initial appearance as separate challenges with seemingly divergent approaches, both of these problems are interconnected components of a more comprehensive problem: "graph motif parameters". In this paper, we provide a precise characterization of the WL-dimension of labeled graph motif parameters. As specific instances of this result, we obtain characterizations of the WL-dimension of the subgraph counting and induced subgraph counting problem for every labeled pattern $P$. Particularly noteworthy is our resolution of a problem left open in previous work concerning induced copies. We additionally demonstrate that in cases where the $k$WL test distinguishes between graphs with varying occurrences of a pattern $P$, the exact number of occurrences of $P$ can be computed uniformly using only local information of the last layer of a corresponding GNN. We finally delve into the challenge of recognizing the WL-dimension of various graph parameters. We give a polynomial time algorithm for determining the WL-dimension of the subgraph counting problem for given pattern $P$, answering an open question from previous work. We additionally show how to utilize deep results from the field of graph motif parameters, together with our characterization, to determine the WL-dimension of induced subgraph counting and counting $k$-graphlets.

## 1 Introduction

**Context.**   Graph neural networks (GNNs) have gained increasing importance over the last decade due to their ability to process and operate over graph-structured data (Wu et al., 2021; Zhou et al., 2020; Joshi et al., 2020). Tasks in which GNNs excel include node classification (Xiao et al., 2022), graph classification (Errica et al., 2020), link prediction (Teru et al., 2020; Zhu et al., 2021), and query answering over knowledge graphs (Daza & Cochez, 2020; Galkin et al., 2022). Based on these, GNNs have found applications in many fields, including social network analysis (Kipf & Welling, 2018), recommender systems (Ying et al., 2018), chemistry (Gilmer et al., 2017), semantic web (Hogan et al., 2022; Schlichtkrull et al., 2018), natural language processing (Marcheggiani & Titov, 2017), and combinatorial optimization (Dai et al., 2021).

The practical importance of GNNs has spurred the community to study their *expressive power*. This refers to the ability of GNNs to distinguish pairs of non-isomorphic graphs. As it was early observed in two landmark articles (Morris et al., 2019; Xu et al., 2019), the expressive power of so-called *message-passing* GNNs (MPGNNs) is precisely that of the Weisfeiler-Leman (WL) test (Weisfeiler & Leman, 1968), one of the most renowned methods for checking graph isomorphism. As recently observed, this correspondence holds even in cases where graphs are node- and edge-labeled, naturally

representing the rich structure present in *knowledge graphs* (Barceló et al., 2022). The above has ignited significant interest among experts in exploring which graph properties can be distinguished by the WL test, especially those that are crucial for the applications of MPGNNs (Arvind et al., 2020; Chen et al., 2020; Morris et al., 2020; Barceló et al., 2021; Huang et al., 2023; Bouritsas et al., 2023).

A class of graph properties that have received special attention in this context is the number of times a given "pattern" appears as a subgraph in a graph. The relevance of this class emanates from the fact that subgraph counts are used in several graph-related tasks, such as constructing graph kernels (Shervashidze et al., 2009; Kriege et al., 2018) and computing spectral graph information (Preciado & Jadbabaie, 2010). Subgraph counts also lie at the basis of some methods that measure the similarity between graphs (Alon et al., 2008). Therefore, it is crucial to comprehend the extent to which these methods align with the expressive capabilities of the WL test (or equivalently, MPGNNs).

In its more general version, the WL test is characterized by a parameterized *dimension*, $k \geq 1$ (Cai et al., 1992; Morris et al., 2019). The $k$-*dimensional* WL test, or $k$WL test for short, iteratively colors the $k$-tuples of nodes in a graph until a fixpoint is reached. Two graphs are said to be *distinguishable* by $k$WL, if the multisets of colors of all $k$-tuples reached at this fixpoint are different. A graph property $f$ can be *distinguished* by $k$WL if for any two graphs $G, H$ where $f(G) \neq f(H)$ only if $G$ and $H$ are distinguishable by $k$WL. We are then interested in the least $k$ for which a parameter can be distinguished by $k$WL. We refer to this $k$ as the *WL-dimension* of the parameter.

Several important results have been obtained over the last few years regarding the WL-dimension of counting the number of copies of a graph $P$ (the *pattern*). We summarize some of such results next, by distinguishing the case in which we count copies of $P$ (subgraphs) from the one in which we count *induced* copies of $P$ (induced subgraphs). For the sake of clarity, we call the former the *subgraph* WL-dimension of $P$ and the latter the *induced subgraph* WL-dimension of $P$.

- *Counting subgraphs:* An important notion in graph theory is the *treewidth* of a graph, which intuitively measures its degree of acyclicity (Diestel, 2012). The *hereditary treewidth* of a pattern $P$ is, in broad terms, the largest treewidth of any of the homomorphic images of $P$. Arvind et al. (2020) initiated the study of the ability of the WL test to count subgraphs in terms of the notion of hereditary treewidth. They established that if the pattern $P$ has hereditary treewidth $k$, then $P$ has subgraph WL-dimension at most $k$. Nevertheless, the sole instance in which they were able to demonstrate the validity of the converse was for $k = 1$, i.e., they showed that $P$ has subgraph WL-dimension one iff $P$ has hereditary treewidth one. In the meantime, some partial results were obtained for the case when $k = 2$ over particular classes of graphs. For instance, by combining results in Arvind et al. (2020) and Fürer (2017), one obtains that the largest cycle (respectively, path) with a subgraph WL-dimension of two is that of length seven. Very recently, however, this gap has been closed. In fact, Neuen (2023) proves the following for each $k \geq 1$: if $P$ is a pattern, then $P$ has subgraph WL-dimension $k$ iff $P$ has hereditary treewidth $k$. This also provides an alternative explanation for the aforementioned result on cycles (resp., paths), as one can observe that a cycle (resp., path) has hereditary treewidth two iff it is of length at most seven.
- *Counting induced subgraphs:* Most of the existing results on counting induced subgraphs were obtained in Chen et al. (2020). The authors show that all patterns with $k + 1$ nodes have an induced subgraph WL-dimension bounded by $k$. Moreover, this is optimal for $k = 1$; i.e., no pattern with three or more nodes has induced subgraph WL-dimension one. It is not known if this correspondence continues to hold for $k > 1$, i.e., whether there are patterns with $k + 2$ or more nodes with induced subgraph WL-dimension $k$, for $k > 1$.

It is noteworthy that the previously mentioned results regarding counting induced subgraphs were achieved in a broader context compared to the results concerning counting subgraphs. Specifically, the former apply even to labeled graphs, where each node and edge is assigned a label, whereas the latter were derived for non-labeled graphs.

Therefore, we have different levels of understanding of the capabilities of the $k$WL test for counting subgraphs and induced subgraphs. Furthermore, these two issues have been addressed separately and employing distinct techniques, which enhances the perception that there is no inherent structural linkage between the two. However, as evidenced by research in the field of counting complexity, there exists a cohesive approach through which these types of problems can be examined. In fact, the counting of subgraphs and induced subgraphs are fundamentally interconnected, akin to opposite

faces of a coin, as they can be represented as linear combinations of one another (Lovász, 1967). Thus, we can achieve insight into either of them by exploring the linear combinations of the other.

Expanding upon this idea, Curticapean et al. (2017) introduced a comprehensive framework for *graph motif parameters*, which are defined as linear combinations of subgraph counts. In this paper, we study the ability of the WL test to count graph motifs, which provides us with a general approach for studying problems related to subgraph counting in this setting. Our main contributions are summarized below. It is worth noting that all such results are derived within the framework established in Chen et al. (2020), which focuses on labeled graphs. This introduces an additional level of intricacy into all the proofs presented in this paper.

- By building on tools developed by Neuen (2023) and Seppelt (2023), we establish a precise characterization for the class of labeled graph motifs with WL-dimension $k$, for $k \geq 1$. Specifically, for subgraph counting, this class precisely corresponds to the patterns of hereditary treewidth $k$, aligning with the characterization presented by Neuen (2023) for the case of unlabeled graphs. For induced subgraph counting, this class contains precisely the patterns featuring $k + 1$ nodes, thus resolving the open issue posed by Chen et al. (2020).

- The previous result characterizes for which graph motifs $\Gamma$ the $k$WL test is able to distinguish between graphs with different numbers of occurrences of $\Gamma$. A natural question arises: Is it possible to obtain the number of occurrences of a graph motif $\Gamma$ in a graph $G$ by computing a function over the multiset of colors of $k$-tuples of vertices obtained from the $k$WL test? We answer this question affirmatively. This result can be of interest to researchers working on MPGNN applications, as it suggests that by designing a suitable MPGNN architecture, one might be able to count the number of subgraphs that appear in a given graph.

- We finally move into the problem of determining the WL-dimension for the problem of counting the occurrences of a given graph pattern $P$. Our characterization shows that for counting induced subgraphs this problem is trivial as the WL-dimension is precisely the number of vertices in $P$ minus 1. For subgraph counting, in turn, the problem is nontrivial, as we have to check for each homomorphic image of $P$ whether its treewidth is at most $k$. Since the number of homomorphic images of $P$ is potentially exponential, this yields a naïve exponential time algorithm for the problem. We show that, in spite of this, the problem admits a polynomial time algorithm. The existence of such an algorithm was left open in Arvind et al. (2020) even for the case $k = 2$.

Since many of the proofs in the paper are extensive and complex, we have opted to place technical details in the appendix and offer proof sketches in the main body of the paper to conserve space.

## 2 PRELIMINARIES

**Labeled graphs.** We work with graphs that contain no self-loops and are node- and edge-labeled. Let $\Sigma$ and $\Delta$ be finite alphabets containing node and edge labels, respectively. A *labelled graph* is a tuple $G = (V, E, \lambda, \kappa)$, where $V$ is a finite set of nodes, $E$ is a set of undirected edges, i.e., unordered pairs of nodes, $\lambda : V \to \Sigma$ is a function such that $\lambda(v)$ represents the label of node $v$, for each $v \in V$, and $\kappa : E \to \Delta$ is a function such that $\kappa(e)$ represents the label of edge $e$, for each $e \in E$.

Given labeled graphs $G = (V, E, \lambda, \kappa)$ and $G' = (V', E', \lambda', \kappa')$, a *homomorphism* from $G$ to $G'$ is a function $f : V \to V'$ such that: (1) $(u, v) \in E \Rightarrow (f(u), f(v)) \in E'$, for each $u, v \in V$, (2) $\lambda(v) = \lambda'(f(v))$, for each $v \in V$, and (3) for each edge $(u, v) \in E$, it holds that $\kappa(u, v) = \kappa'(f(u), f(v))$. If, in addition, $f$ is a bijection and the first condition is satisfied by the stronger statement that $(u, v) \in E \Leftrightarrow (f(u), f(v)) \in E'$, for each $u, v \in V$, then we say that $f$ is an *isomorphism*. We write $\mathsf{Hom}(G, G')$ for the set of all homomorphisms from $G$ to $G'$ and $\mathsf{homs}(G, G')$ for the number of homomorphisms in $\mathsf{Hom}(G, G')$.

For a $k$-tuple $\bar{v} = (v_1, \ldots, v_k) \in V(G)^k$, we write $G[\bar{v}]$ as a shortcut for $G[\{v_1, \ldots, v_k\}]$. The *atomic type* $\mathsf{atp}(G, \bar{v})$ of a $k$-tuple $\bar{v}$ of vertices in a labeled graph $G$ is some function such that, for $\bar{v} \in V(G)^k$ and $\bar{w} \in V(H)^k$, it holds that $\mathsf{atp}(G, \bar{v}) = \mathsf{atp}(H, \bar{w})$ if and only if the mapping $v_i \mapsto w_i$ is an isomorphism from $G[\bar{v}]$ into $H[\bar{w}]$.

**Weisfeiler-Leman test.** Let $G$ be a labeled graph. The $k$WL test, for $k > 0$, iteratively colors the elements in $V(G)^k$, that is, all tuples of $k$ nodes from $V(G)$. The color of tuple $\bar{v} \in V(G)^k$ after $i$ iterations, for $i \geq 0$, is denoted $c_i^k(\bar{v})$. This coloring is defined inductively, and its definition depends on whether $k = 1$ or $k > 1$.

- For $k = 1$, we have that

$$c_i^k(v) := \begin{cases} \mathsf{atp}(G, v) & \text{if } i = 0 \\ \left(c_{i-1}^k(v), \{\!\{(\kappa(v, w), c_{i-1}^k(w)) \mid (v, w) \in V(G)\}\!\}\right) & \text{if } i \geq 1 \end{cases}$$

In other words, $c_i^k(v)$ consists of the color of vertex $v$ in the previous iteration, along with a multiset that includes, for each neighbor $w$ of $v$ in $G$, an ordered pair consisting of the color of $w$ in the previous iteration and the label of the edge connecting $v$ and $w$.

- For $k > 1$, we have that

$$c_i^k(\bar{v}) := \begin{cases} \mathsf{atp}(G, \bar{v}) & \text{if } i = 0 \\ \left(c_{i-1}^k(\bar{v}), \{\!\{\mathsf{ct}(w, i-1, \bar{v}) \mid w \in V(G)\}\!\}\right) & \text{if } i \geq 1 \end{cases}$$

Here, $\mathsf{ct}(w, i, \bar{v})$ denotes the *color tuple* for $w \in V(G)$ and $\bar{v} \in V(G)^k$, and is defined as:

$$\mathsf{ct}(w, i, \bar{v}) = \left(c_i^k(\bar{v}[w/1]), \dots, c_i^k(\bar{v}[w/k])\right),$$

where $\bar{v}[w/j]$ denotes the tuple that is obtained from $\bar{v}$ by replacing its $j$th component with the element $w$. In other words, $c_i^k(\bar{v})$ is formed by the color of $\bar{v}$ in the previous iteration, along with a multiset that includes, for each node $w$ in $G$, a tuple containing the color in the previous iteration for each tuple that can be derived from $\bar{v}$ by substituting one of its components with the element $w$.

The $k$WL test *stabilizes* after finitely many steps. That is, for every labeled graph $G$ there exists a $t \geq 0$ such that, for each $\bar{v}, \bar{w} \in V(G)^k$, it holds that

$$c_t^k(\bar{v}) = c_t^k(\bar{w}) \iff c_{t+1}^k(\bar{v}) = c_{t+1}^k(\bar{w}).$$

We then define the *color of tuple $\bar{v}$ in $G$* as $c^k(\bar{v}) := c_t^k(\bar{v})$.

We say that two labeled graphs $G$ and $H$ are *indistinguishable* by $k$WL (or $G \equiv_{k\text{WL}} H$), if the $k$-WL algorithm yields the same coloring on both graphs, i.e.,

$$\{\!\{c^k(\bar{v}) \mid \bar{v} \in V(G)^k)\}\!\} = \{\!\{c^k(\bar{w}) \mid \bar{w} \in V(H)^k)\}\!\}.$$

The version of the WL test used in this paper is also known as the *folklore $k$WL test* (as defined, for instance, by Cai et al. (1992)). It is essential to note that an alternative version of this test, known as the *oblivious $k$WL test*, has also been explored in the machine learning field (Morris et al., 2019). Notably, it is established that for each $k \geq 1$, the folklore $k$WL test possesses the same distinguishing capability as the oblivious $(k + 1)$WL test (Grohe & Otto, 2015).

**Graph motif parameters.** A *labeled graph parameter* is a function that maps labeled graphs into $\mathbb{Q}$. A *labeled graph motif parameter* (Curticapean et al., 2017) is a labeled graph parameter such that, for a labeled graph $G$, its value in $G$ is equivalent to a (finite) linear combination of homomorphism counts into $G$. More formally, function $\Gamma$ is a labeled graph motif parameter, if there are fixed labeled graphs $F_1, \dots, F_\ell$ and constants $\mu_1, \dots, \mu_\ell \in \mathbb{Q} \setminus \{0\}$, such that for every labeled graph $G$:

$$\Gamma(G) = \sum_{i=1}^{\ell} \mu_i \mathsf{homs}(F_i, G). \tag{1}$$

We refer to the set $\{F_1, \dots, F_\ell\}$ as the *support* $\mathsf{Supp}(\Gamma)$ of $\Gamma$.

It is often not easy to see whether a function on graphs is a graph motif parameter. Fortunately, we know that certain model counting problems for large fragments of first-order logic are indeed labeled graph motif parameters. In the following, we view labeled graphs as structures. Logical formulas can then be defined over a signature that contains a unary relation symbol $U_\sigma$, for each label $\sigma \in \Sigma$, and a binary relation symbol $E_\delta$, for each label $\delta \in \Delta$. Let $\varphi$ be a formula with free variables $\bar{x}$ of the form

$$\exists \bar{y} \ \psi(\bar{y}, \bar{x}) \wedge \psi^*(\bar{x}),$$

where $\psi$ is a formula consisting of positive atoms, conjunction and disjunction, and $\psi^*$ is a conjunction of inequalities and possibly negated atoms. We call such formulas *positive formulas with free constraints*. We write $\#\varphi(G)$ for the number of assignments to $\bar{x}$ for which the formula is satisfied in labeled graph $G$. It is known that the function $\#\varphi$ for such formulas can be expressed as a finite linear combination of homomorphism counts *with projection*. Göbel et al. (2024) have recently initiated the study of how homomorphism counts with projection relate to the $k$WL test. In this paper, we are content with a more restricted fragment for which no projection in the homomorphism counts is required.

**Proposition 1** (Chen & Mengel (2016),Dell et al. (2019))**.** *Let $\varphi$ be a quantifier-free positive formula with free constraints. Then the function $\#\varphi$ is a labeled graph motif parameter[1].*

**Example 1.** We start by showing that counting subgraphs and counting induced subgraphs are examples of graph motif parameters (for each fixed pattern). Suppose for simplicity that we consider graphs without node labels and with a single edge label, and let $E$ be the corresponding relation symbol. Consider a graph $H = (V, E)$ for which we want to count occurrences as a subgraph. We can do this via the following positive formula with free constraints:

$$\phi_H := \bigwedge_{(u,v) \in E} E(x_u, x_v) \wedge \bigwedge_{u \neq v,\, u,v \in V} x_u \neq x_v.$$

Notice that this formula has a free variable $x_v$, for each node $v \in V$. If now we want to count the number of occurrences of $H$ as an induced subgraph, we can simply extend $\phi_H$ with the following conjunction: $\bigwedge_{(u,v) \notin E} \neg E(x_u, x_v)$. graph motif parameter by $\mathsf{ind}_H$.

Consider the following formula with free variables $x_1, \ldots, x_k$:

$$\phi_{\mathrm{IS}} := \bigwedge_{1 \leq i \neq j \leq k} \neg E(x_i, x_j)$$

The satisfying assignments to $\phi_{\mathrm{IS}}$ in graph $G$ are precisely its independent sets of size at most $k$. We see then that counting $k$-independent sets is a graph motif parameter. $\qquad\square$

**Counting graph motifs parameters and the WL test.**    Take $k > 0$. Consider a graph parameter $\Gamma$. We state that the $k$WL test *can distinguish* $\Gamma$ if, for any pair of labeled graphs $G$ and $H$ where $\Gamma(G) \neq \Gamma(H)$, it follows that $G \not\equiv_{k\mathrm{WL}} H$. The *WL-dimension* of a graph parameter $\Gamma$ is the minimal $k > 0$ such that the $k$WL test can distinguish $\Gamma$.

## 3    CHARACTERIZING THE WL-DIMENSION OF GRAPH MOTIF PATTERNS

In this section, we provide a characterization of the WL-dimension of labeled graph motif parameters. To grasp the results presented in this section, it is crucial to first define the concept of the *treewidth* of a graph $G$. This concept, well-known in graph theory, aims to quantify the degree of acyclicity in $G$.

Let $G = (V, E, \lambda, \kappa)$ be a labeled graph. A *tree decomposition* of $G$ is a tuple $(T, \alpha)$, where $T$ is a tree and $\alpha$ is a function that maps each node $t$ of the tree $T$ to a subset of the nodes in $V$, that satisfies the following:

- For each $(u, v) \in E$, there exists a node $t \in T$ with $\{u, v\} \in \alpha(t)$.
- For each node $v \in V$, the set $\{t \in T \mid v \in \alpha(t)\}$ is a subtree of $T$. In other words, the nodes $t$ of $T$ for which $\alpha(t)$ contains $v$ are connected in $T$.

The *width* of a tree decomposition $(T, \alpha)$ of $G$ is defined as $\max_{t \in T} |\alpha(t)| - 1$. The treewidth of $G$ is then defined as the minimum width of any of its tree decompositions. It is easy to see that $G$ has treewidth one if and only if its underlying graph is a tree.

The following is the main result in this section. It characterizes the WL-dimension of a graph motif parameter in terms of the treewidth of its support set.

**Theorem 2.** *Let $\Gamma$ be a labeled graph motif parameter. The WL-dimension of $\Gamma$ is the maximum treewidth of any labeled graph in $\mathsf{Supp}(\Gamma)$.*

---

[1]Dell et al. (2019) additionally require $\psi$ to be in CNF/DNF. For our purposes the potential blowup incurred by transformation into CNF is of no consequence.

Before providing a proof sketch of this result, we show some examples of how the theorem can be applied to concrete instances of labeled graph motif patterns used in this paper.

**Example 2.** Let us consider first the case of counting subgraphs for a labeled graph $H$. We denote the corresponding labeled graph motif pattern by $\mathsf{Sub}_H$. We assume that $H$ does not have self-loops. We call a labeled graph $H'$ a *homomorphic image* of $H$, if there exists a surjective homomorphism from $H$ into $H'$. Notice that the set of homomorphic images of $H$ contains a finite number of labeled graphs up to isomorphism. Following Curticapean et al. (2017), we denote by $\mathsf{spasm}(H)$ the set of all labeled graphs that are obtained from the homomorphic images of $H$ by removing self-loops. As shown in Curticapean et al. (2017), the support of $\mathsf{Sub}_H$ is precisely the set $\mathsf{spasm}(H)$. It follows from Theorem 2 that the $H$s for which $k$WL can distinguish subgraph counting for $\mathsf{Sub}_H$ are precisely those for which the maximum treewidth of a labeled graph in $\mathsf{spasm}(H)$ is at most $k$. This result has been established for unlabeled graphs in Neuen (2023). Here, we extend this result to labeled graphs.

Consider now the case of counting induced subgraphs for $H$. We write $\mathsf{Ind}_H$ for the corresponding labeled graph motif pattern It can be shown that, in this case, the support for $\mathsf{Ind}_H$ is the set of all labeled graphs $H'$ that can be obtained from $H$ by adding edges among its nodes. If $H$ has $k$ nodes, then the support for $H$ contains the clique of size $k$ which is known to have treewidth $k - 1$. Furthermore, this is the labeled graph with the largest treewidth in the support of $H$. It follows from Theorem 2 that the $H$s for which $k$WL can distinguish induced subgraph counting for $\mathsf{Ind}_H$ are precisely those with at most $k + 1$ nodes. This solves an open question from Chen et al. (2020). $\qquad\square$

We now discuss our proof of Theorem 2. The full argument requires us to extend various existing results in the field to the setting of labeled graphs. We defer those proofs to the appendix. Let $\mathcal{F}$ be a class of labeled graphs. We write $G \equiv_{\mathcal{F}} H$ if for every $F \in \mathcal{F}$ it holds that $\mathsf{homs}(F, G) = \mathsf{homs}(F, H)$. Following Roberson (2022), we define the *labeled homomorphism-distinguishing closure* of $\mathcal{F}$, denoted $\mathsf{homclosure}(\mathcal{F})$, as the set of all labeled graphs $L$ for which the following holds for every labeled graphs $G, H$:

$$G \equiv_{\mathcal{F}} H \implies \mathsf{homs}(L, G) = \mathsf{homs}(L, H).$$

We say that $\mathcal{F}$ is *labeled homomorphism-distinguishing closed*, if $\mathsf{homclosure}(\mathcal{F}) = \mathcal{F}$. A recent paper by Neuen establishes that the class of all (unlabeled) graphs of treewidth at most $k$ is labeled homomorphism-distinguishing closed (Neuen, 2023). As we establish next, this extends to the labeled setting. Here we denote by $\mathcal{LT}_k$ the class of all labeled graphs of treewidth at most $k$.

**Lemma 3.** *Fix $k > 0$. The class $\mathcal{LT}_k$ is labeled homomorphism-distinguishing closed.*

*Proof Sketch.* For our proof we show that a recent breakthrough result by Roberson (2022) holds also in the labeled case. The result depends on the notion of *weak oddomorphisms* which, roughly speaking, are a particular kind of homomorphism that satisfies additional constraints (see Appendix B for details). To simplify the presentation, we actually only show that Roberson's result holds for *edge-labeled* graphs, i.e., those labeled graphs where all vertices have the same label. We then show that this is enough and that we can lift the homomorphism-distinguishing closedness from the edge-labeled case to the general labeled case.

Formally, we show the labeled analogue of (Roberson, 2022, Theorem 6.2). Namely, that every class of labeled graphs $\mathcal{F}$ that satisfies the following two closure properties is also labeled homomorphism-distinguishing closed. (1) If $F \in \mathcal{F}$, and $F$ has a weak oddomorphism into $G$, then $G \in \mathcal{F}$. (2) $\mathcal{F}$ is closed under restriction to connected components and disjoint union. Once we have established that the theorem still holds, we show that both properties are satisfied by $\mathcal{LT}_k$. For the first, in particular, we can build on parts of Neuen's argument for the unlabeled graph case. The second property is a well known property of treewidth and requires no further insights. $\qquad\square$

The second intermediate result that we need is the following lemma, proved recently by Seppelt.

**Lemma 4** (Lemma 4 in (Seppelt, 2023))**.** *Let $\mathcal{F}$ and $\mathcal{L}$ be classes of labeled graphs. Suppose $\mathcal{F}$ is finite and its elements are pairwise non-isomorphic. For each $F \in \mathcal{F}$, let $\mu_L$ be a element of $\mathbb{R}$. Then $\mathcal{F} \subseteq \mathsf{homclosure}(\mathcal{L})$, if for all labeled graphs $G, H$ we have that:*

$$G \equiv_{\mathcal{L}} H \implies \sum_{F \in \mathcal{F}} \mu_F \cdot \mathsf{homs}(F, G) = \sum_{F \in \mathcal{F}} \mu_F \cdot \mathsf{homs}(F, H).$$

We also need the following result which establishes that two labeled graphs $G$ and $H$ are indistinguishable by $k$WL if and only if $\mathsf{homs}(F,G) = \mathsf{homs}(F,H)$, for every labeled graph $F$ of treewidth $k$. The unlabeled analogue of this result is due to Dvořák (2010) (rediscovered by Dell et al. (2018)).

**Lemma 5.** *For all labeled graphs $G, H$ we have $G \equiv_{kWL} H$ if and only if $G \equiv_{\mathcal{LT}_k} H$.*

*Proof of Theorem 2.* Let $\mathcal{F}$ be the support of $\Gamma$ and $k$ be the maximum treewidth of a labeled graph in $\mathcal{F}$, for $k > 0$. We first show that $k$WL can distinguish $\Gamma$. Take two labeled graphs $G, H$ with $G \equiv_{kWL} H$. By Lemma 5, we also have that $G \equiv_{\mathcal{LT}_k} H$. In particular, $\mathsf{homs}(F,G) = \mathsf{homs}(F,H)$, for every $F \in \mathcal{F}$, and therefore:

$$\Gamma(G) \;=\; \sum_{F \in \mathcal{F}} \mu_F \cdot \mathsf{homs}(F,G) \;=\; \sum_{F \in \mathcal{F}} \mu_F \cdot \mathsf{homs}(F,H) \;=\; \Gamma(H).$$

Suppose now, for the sake of contradiction, that the $\ell$WL test can distinguish $\Gamma$, for $\ell < k$. Let $G, H$ be arbitrary labeled graphs. Notice that we have the following:

$$G \equiv_{\mathcal{LT}_\ell} H \;\Rightarrow\; G \equiv_{\ell WL} H \;\Rightarrow\; \Gamma(G) = \Gamma(H) \;\Rightarrow\; \sum_{F \in \mathcal{F}} \mu_F \cdot \mathsf{homs}(F,G) = \sum_{F \in \mathcal{F}} \mu_F \cdot \mathsf{homs}(F,H).$$

The first implication follows from Lemma 5 and the second one since the $\ell$WL test can distinguish $\Gamma$. But then Lemma 4 tells us that $\mathcal{F} \subseteq \mathsf{homclosure}(\mathcal{LT}_\ell)$, and Lemma 3 that $\mathsf{homclosure}(\mathcal{LT}_\ell) = \mathcal{LT}_\ell$. This is a contradiction since $\mathcal{F}$ contains at least one labeled graph of treewidth $k > \ell$. $\square$

## 4 Counting occurrences of graph motif patterns

In this section, we establish that if a graph motif parameter $\Gamma$ has WL-dimension $k$, then one can actually obtain the number of occurrences of $\Gamma$ in a graph $G$ by looking independently only at the colors of the individual $k$-tuples, rather than the full multiset of stable colors for all $k$-tuples. This is of particular interest when looking at the $k$WL test from the perspective of MPGNNs. The natural expression of the $k$WL test in MPGNNs leads to a final layer that assigns a color to each $k$-tuple of vertices. So, if $\Gamma$ is a labeled graph motif parameter, and we want to compute $\Gamma(G)$ over a labeled graph $G$, then Theorem 6 shows that it is not necessary to combine the information of the final layer in a global way, but that there is some uniform function that can map each individual color to a rational number, the sum of which will exactly be $\Gamma(G)$.

More formally, we show the following result.

**Theorem 6.** *Let $\Gamma$ be a labeled graph motif parameter and suppose that the maximum treewidth of a labeled graph in $\mathsf{Supp}(\Gamma)$ is at most $k$. Also, let $\mathcal{C}^k$ denote the set of possible colors produced by the $k$WL test. Then there exists a function $\theta_\Gamma : \mathcal{C}^k \to \mathbb{Q}$ such that $\Gamma(G) = \sum_{\bar{v} \in V(G)^k} \theta_\Gamma(c^k(\bar{v}))$, for every labeled graph $G$.*

Theorem 6 is obtained by using the following lemma, whose proof can be found in the appendix.

**Lemma 7.** *Let $F$ be a labeled graph with treewidth at most $k$ and let $\mathcal{C}^k$ denote the set of possible colors produced by the $k$WL test. There exists a function $\eta_F : \mathcal{C}^k \to \mathbb{N}$ such that $\mathsf{homs}(F,G) = \sum_{\bar{v} \in V(G)^k} \eta_F(c^k(\bar{v}))$, for each labeled graph $G$.*

Theorem 6 is then obtained from Lemma 7 as follows. Let us assume that $\mathsf{Supp}_\Gamma = \{F_1, \ldots, F_\ell\}$ and $\Gamma(G)$ is defined as in Equation (1). We define $\theta_\Gamma : \mathcal{C}^k \to \mathbb{Q}$ as $\theta_\Gamma(c) = \sum_{i=1}^{\ell} \mu_i \cdot \eta_{F_i}(c)$, for each $c \in \mathcal{C}^k$. We then have that

$$\begin{aligned}
\Gamma(G) \;&=\; \sum_{i=1}^{\ell} \mu_i \cdot \mathsf{homs}(F_i, G) && \text{(Equation (1)} \\[2mm]
&=\; \sum_{i=1}^{\ell} \left( \sum_{\bar{v} \in V(G)^k} \mu_i \cdot \eta_{F_i}(c^k_{\bar{v}}) \right) && \text{(Lemma 7)} \\[2mm]
&=\; \sum_{\bar{v} \in V(G)^k} \left( \sum_{i=1}^{\ell} \mu_i \cdot \eta_{F_i}(c^k_{\bar{v}}) \right) \;=\; \sum_{\bar{v} \in V(G)^k} \theta_\Gamma(c^k_{\bar{v}}).
\end{aligned}$$

## 5 DETERMINING THE WL-DIMENSION FOR SUBGRAPH COUNTING

We finally move on the recognizability problem for WL-dimension. That is, for given graph motif parameter $\Gamma$, we want to know the WL-dimension of $\Gamma$. Arvind et al. (2020) previously raised the question for which labeled graph patterns $H$, the graph parameter $\mathsf{Sub}_H$ can be expressed by 2WL. Here, we give a polynomial time algorithm that decides the WL-dimension for all subgraph counting problems, thus also resolving this question as a special case. For the analogous problem $\mathsf{Ind}_H$, we have already seen in Example 2 that the recognition problem is, in fact, trivial. In addition, we determine the WL-dimension of counting $k$-*graphlets*, i.e., all connected induced subgraphs on $k$ vertices, as illustrative examples of how further such results can be derived from the deep body of work in graph motif parameters.

We will begin with the case of subgraph counting. Recall that a *minor* of a labeled graph $H$ is a labeled graph $H'$ that can be obtained from $H$ by removing nodes or edges, or contracting edges (that is, removing an edge and simultaneously merging its endpoints). By Theorem 2, and the following discussion on $\mathsf{spasm}(H)$, recognising the WL-dimension of $\mathsf{Sub}_H$ is precisely the problem of recognising the maximum treewidth of the homomorphic images of $H$. We can use classic results in the field of graph minors to express this as a property checkable in *monadic second-order logic* (MSO), for which model checking is tractable in our setting. We give a more detailed sketch of the argument below, full details are provided in the appendix.

**Theorem 8.** *Fix $k > 0$. There is a polynomial time algorithm for the following problem: given a labeled graph $H$, checking if the WL-dimension of $\mathsf{Sub}_H$ is at most $k$.*

*Proof Sketch.* The algorithm first checks whether the treewidth of $H$ is at most $k$. It is well known that this can be decided in linear time for fixed $k$ Bodlaender (1996). Obviously, $H \in \mathsf{spasm}(H)$, and thus if the treewidth of $H$ is strictly larger than $k$, we are done and the algorithm rejects. By the Robertson-Seymour Theorem (Robertson & Seymour, 2004), there is a finite set of graphs $\mathcal{F}_k$, such that there is a $F \in \mathcal{F}_k$ that is a minor of graph $G$ if and only if the treewidth of $H$ is at most $k$. We then show that there is an MSO formula $\varphi_F$ such that

$$H \vDash \varphi_F \iff F \text{ is a minor of some homomorphic image of } H.^2$$

Clearly then $\varphi = \bigvee_{F \in \mathcal{F}_k} \varphi_k$ is an MSO formula such that $H \vDash \varphi$ if and only if the maximum treewidth in the spasm is at most $k$. By a standard algorithmic metatheorem of Courcelle (Courcelle, 1990), deciding $H \vDash \varphi$ is possible in time $f(\ell, \varphi)|H|$, where $\ell$ is the treewidth of $H$. As $\ell \le k$ by our initial check, and $\varphi$ depends only on $k$, the problem is in $O(|H|)$ for fixed $k$. $\square$

We now move on to study the WL-dimension of counting $k$-graphlets, a popular problem in graph mining and analysis of social networks Bressan et al. (2017); Jin et al. (2018). Here, no dedicated algorithm is necessary as the WL-dimension will be immediate from observations about the respective support when viewed as graph motif parameters. While the problem of counting $k$-graphlets is itself popular, its analysis in our context serves as a example of how the recently established theory of graph motif parameters can answer these types of recognizability questions.

The logical view, via positive formulas with free constraints, is not the only natural way to see that a graph parameter is indeed also a graph motif parameter. An alternative notion that is known to correspond to graph motif parameters is the problem of counting all induced subgraphs of size $k$, that satisfy some arbitrary (computable) property $\phi$ (Roth & Schmitt, 2020). Thus, with $\phi$ being the property of the graph being connected, this immediately tells us that counting $k$-graphlets is a graph motif parameter. In fact, Roth & Schmitt (2020) not only shows that every such property is a graph motif parameter, but they also determine precisely when the $k$-clique is in the support (i.e., has a coefficient $\neq 0$). Moreover, in the case of properties that are closed under the removal of edges, there is a strong connection to combinatorial topology that can be leveraged to make this determination.

**Proposition 9.** *For $k > 1$, counting the number of $k$-graphlets has WL-dimension $k - 1$.*

*Proof Sketch.* Let $\mathsf{Ind}_k^{\mathsf{C}}$ be graph parameter that counts the number of induced subgraphs on $k$ vertices that are connected, i.e., the number of $k$-graphlets. For graph property $\phi$, let $E_k^\phi$ be the set of all

---

[2]Alternatively, it is also possible to show that the maximal treewidth in $\mathsf{spasm}_H$ is minor-closed and obtain a different set of forbidden minors for $H$ itself.

edge-subsets of the complete graph on $k$ vertices, such that the corresponding subgraph satisfies property $\phi$. Roth & Schmitt (2020) showed that the problem of counting all induced subgraphs on $k$ vertices that satisfy $\phi$ is a graph motif parameter, and that the $k$-clique is in the support of the parameter precisely when

$$\sum_{A \in E_k^\phi} (-1)^{|A|} \neq 0.$$

Furthermore, they show that if $\phi$ is closed under removal of edges, then $E_k^\phi \setminus \{0\}$ forms a so-called simplicial graph complex and the reduced Euler characteristic of this simplicial complex is also non-zero exactly when the $k$-clique is in the support of the parameter (see (Jonsson, 2008) for details regarding these notions).

Now, let us use C for the property of being connected and NC for its complement, i.e., disconnectedness. Property NC is closed under removal of edges and its simplicial complexes are well understood (cf., (Jonsson, 2008)). In particular, their reduced Euler characteristic is known to be $\pm(k-1)!$, which is non-zero for natural $k$. Now, it is enough to observe that

$$\sum_{A \in E_k^{\mathsf{NC}}} (-1)^{|A|} + \sum_{A \in E_k^{\mathsf{C}}} (-1)^{|A|} = 0$$

as the properties are complementary and the sums thus form an alternating sum of binomials. Since the left-hand sum is non-zero, so is the right-hand sum for property C. We can therefore conclude that the $k$-clique is in the support of $\mathsf{Ind}_k^{\mathsf{C}}$[3], demonstrating that the maximal treewidth in the support is at least $k-1$. The argument of Roth & Schmitt (2020) also implicitly shows by construction that the support contains no graphs with more than $k$ vertices, thus also confirming that the treewidth in $\mathsf{Supp}(\mathsf{Ind}_k^{\mathsf{C}})$ is no higher than $k-1$. Applying Theorem 2 completes the argument. □

Moreover, Proposition 1 is constructive and in many cases it is feasible to determine the support from logical formulations of parameters as in Example 1. For example, following the construction of Dell et al. (2019) for the formula $\phi_{IS}$ reveals that counting $k$-independent sets also has WL-dimension $k-1$. What is particularly noteworthy in this context, is that we can use the exact same methods that have been used to study the parameterized complexity of these problems. Just as the WL-dimension, the complexity of computing graph motif parameter $\Gamma$ depends precisely on the maximum treewidth in $\mathsf{Supp}(\Gamma)$ (Curticapean et al., 2017). That is, analysis of the complexity of computing a graph motif parameter and analysis of the WL-dimension of the parameter revolve around the same question of how high the treewidth of the basis can become.

## 6 Conclusions and Limitations

We have shown that recent developments in graph theory and counting complexity can be brought together to provide a precise characterization of the WL-dimension of labeled graph motif parameters. We have also shown that if a graph motif parameter $\Gamma$ belongs to such a class, then the number of "appearances" of $\Gamma$ in a given labeled graph $G$ can be computed uniformly only from local information about the individual $k$-tuples from the output of the $k$WL test. Based on known results this suggests that $k$-dimensional MPGNNs are capable of computing this number, if equipped with the right pooling functions (Morris et al., 2019; Xu et al., 2019). Some work on this direction has already been carried out in Chen et al. (2020) by using *local relational polling*. Finally, we have used our characterization to show that both the classes of patterns for which the $k$WL test can distinguish labeled subgraph counting and labeled induced subgraph counting can be recognized in polynomial time.

The main limitation of our work is that it only concerns the worst-case behavior of the algorithms considered. In fact, the counterexamples our proof constructs for cases in which the $k$WL test is not capable of counting the number of occurrences of a graph motif $\Gamma$ are rather complicated and do not necessarily resemble the cases encountered in practice. Therefore, it would be interesting to understand to what extent these results relate to average-case scenarios, or how well they align with practical applications. We leave this for future work.

---

[3]This observation was already made implicitly but without details in Roth (2019).

## ACKNOWLEDGEMENTS

We are very grateful to Marc Roth for many valuable and enjoyable discussions that have influenced this work. Matthias Lanzinger acknowledges support by the Royal Society "RAISON DATA" project (Reference No. RP\R1\201074) and by the Vienna Science and Technology Fund (WWTF) [10.47379/ICT2201]. Pablo Barceló has been funded by the National Center for Artificial Intelligence CENIA FB210017, Basal ANID, and by ANID Millennium Science Initiative Program Code ICN17002.

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

# A    PROOF OF LEMMA 7

This section focuses on the proof of Lemma 7. While it is well known that the $k$WL test of a graph $G$ determines the homomorphism counts from every graph $H$ with $tw(H) < k$, we are interested in a slightly stronger statement. In particular, let $H$ be a graph and let $\bar{a}$ be some tuple of $k$ distinct vertices in $H$. We show that any two $k$-tuples of vertices $\bar{v}, \bar{w}$, possibly from different graphs, that obtain the same stable color by the $k$WL test, there are the same number of homomorphisms extending $\bar{a} \mapsto \bar{v}$ and $\bar{a} \mapsto \bar{w}$ to the respective graphs of $\bar{v}$ and $\bar{w}$. As a consequence, we see that it is not necessary to know the full multiset of stable colors to determine the homomorphism count from some pattern $F$, but rather $\mathsf{homs}(F, \cdot)$ can be computed by computing a partial count for each $k$-tuple independently and simply summing them up.

This is of particular interest when looking at the $k$WL test from the perspective of GNNs. The natural expression of the $k$WL test in MPGNNs leads to a final layer that assigns a color to each $k$-tuple of vertices. If we want to compute $\mathsf{homs}(F, \cdot)$ with a GNN, Lemma 7 shows that it is not necessary to combine the information of the final layer in some global way, but that there is some uniform function that can map each individual color to a rational number, the sum of which will exactly be $\mathsf{homs}(F, \cdot)$.

We proceed with some technical definitions. For tree decomposition $(T, B)$ and subtree $T'$ of $T$ we will write $B(T')$ to mean $\bigcup_{u \in V(T')} B(u)$. It will be convenient to consider *nice* tree decompositions. A nice TD of labeled graph $G$ is a rooted TD where the bags of all leaves are singletons and all non-root nodes are only of three types:

- An *introduce node* $u$ is the only child of parent $p$, and has $B(u) = B(p) \cup \{a\}$ for some $a \in V(G) \smallsetminus B(p)$.
- A *forget node* $u$ is the only child of parent $p$, and has $B(u) = B(p) \smallsetminus \{a\}$ for some $a \in B(p)$.
- *Split nodes* $u, v$ are the only two children of parent $p$, and have $B(u) = B(v) = B(p)$.

It is well known that if a graph has treewidth $k$, then it also has a nice TD of width $k$.

For labeled graphs $F, G$ with $\bar{a} \in V(F)^k, \bar{v} \in V(G)^k$ we write $\mathsf{Hom}(F, G)[\bar{a} \mapsto \bar{v}]$ for the set of all homomorphisms from $F$ into $G$ that extend the mapping $\bar{a} \mapsto \bar{v}$. For a vertex $v$ of labeled graph $F$ we write $F - v$ to mean $F[V(F) \smallsetminus \{v\}]$.

**Proposition 10.** *Let $(T, B)$ be a tree decomposition of labeled graph $F$. Let $T'$ be a subtree of $T$ and let $\alpha$ be a node adjacent to $T'$ in $T$. For any $z \in B(\alpha) \smallsetminus B(T')$ and $y \in B(T')$ it holds that if $\{y, z\} \in E(F)$, then $y \in B(\alpha)$.*

*Proof.* Suppose the statement were false. Then there is an edge $\{y, z\}$ with $z$ in $B(\alpha)$ but not in $B(T')$, and $y \notin B(\alpha)$. Let $T^*$ be the subtree of $T \smallsetminus T'$ that contains $\alpha$. By connectedness, $z$ occurs only in bags of $T^*$, whereas $y$ occurs in none of them. Hence, the edge $\{y, z\}$ cannot be subset of any bag, contradicting that $(T, B)$ is a tree decomposition. $\square$

Our plan in the proof of Lemma 7 will be to connect nice tree decompositions to $k$WL colors. Intuitively, just as the $k$WL test moves "outward" one vertex at a time at each step, a nice tree decomposition changes only one vertex at a time from parent to child. It is known that $\mathsf{Hom}(F, \cdot)$ is determined by the local homomorphisms from $F[B(u)]$ for each node $u$ of a tree decomposition, together with the overlap of bags in neighboring nodes in the tree decomposition. Roughly speaking, we show that a color produced by the $k$WL test has enough information to determine these two facets.

Our formalization of this idea will require significant technical overhead for bookkeeping. To this end we introduce the following notioons. Let $F, G, H$ be labeled graphs, let $\bar{v} \in V(G)^k$, $\bar{w} \in V(H)^k$ and let $\mu : V(F) \to V(G), \nu : V(F) \to V(H)$. For set $S \subseteq V(F)$ with $|S| \leq k$ we say that $\mu, \nu$ *are in* $(\bar{v}, \bar{w})$-*strict $S$-agreement* if $\mu$ maps vertices in $S$ only to vertices in $\bar{v}$ and $\mu(x) = v_i \iff \nu(x) = w_i$ for all $x \in S$. For a tree $T$ with node labels $L(u) \subseteq V(F)$ of $F$ we say that two homomorphisms are in *colorful strict $T$ leaf agreement* if for every leaf $\ell$ of $T$ there exist $\bar{v} \in V(G)^k, \bar{w} \in V(H)^k$ such that $\mu, \nu$ are in $(\bar{v}, \bar{w})$-strict $L(\ell)$-agreement and for some $i \geq 1$ either $c_i^k(\bar{v}) = c_i^k(\bar{w})$ if $|L(\ell)| \leq k$ or $\mathsf{ct}(a, i, \bar{v}^\ominus) = \mathsf{ct}(b, i, \bar{w}^\ominus)$ where $\bar{v} = \bar{v}^\ominus a, \bar{w} = \bar{w}^\ominus b$ otherwise.

**Lemma 11.** *Let $F$, $G$ and $H$ be labeled graphs, let $\bar{v} \in V(G)^k$, $\bar{w} \in V(H)^k$ with $c_i^k(\bar{v}) = c_i^k(\bar{w})$ for some $i \geq 1$, and let $S \subseteq V(F)$ with $|S| \leq k$. Let $\mu \in \mathsf{Hom}(F, G), \nu \in \mathsf{Hom}(F, H)$ such that they are in*

$(\bar{v}, \bar{w})$-strict $S$-agreement. Let $F'$ be a labeled graph with vertex $x$ such that $F' - x = F$ and $x$ is only adjacent to vertices in $S$. Then there is a bijection $\iota$ between $\mathsf{Hom}(F', G)[\mu]$ and $\mathsf{Hom}(F', H)[\nu]$ such that

1. for all $\mu' \in \mathsf{Hom}(F', G)[\mu]$ and $\iota(\mu')$ there are $\bar{v}' = \bar{v}a, \bar{w}'b$ for $a \in V(G), b \in V(H)$, such that $\mu', \iota(\mu')$ are in $(\bar{v}', \bar{w}')$-strict $S \cup \{x\}$-agreement, and

2. $\mathsf{ct}(a, i - 1, \bar{v}) = \mathsf{ct}(b, i - 1, \bar{w})$.

*Proof.* Let $S' = S \cup \{x\}$. Suppose $\mu' = \mu \cup \{x \mapsto a\}$ is in $Hom(F', G)$. Since $c_i^k(\bar{v}) = c_i^k(\bar{w})$ there must be some $b \in V(H)$ such that $\mathsf{ct}(a, i - 1, \bar{v}) = \mathsf{ct}(b, i - 1, \bar{w})$. Let $\nu' = \nu \cup \{x \mapsto b\}$. Observe that because of the equal color tuples we have in particular, $atp(G, \bar{v}a) = atp(H, \bar{w}b)$. As $x$ is only adjacent to vertices in $S'$ we have that for $\{y, x\} \in E(F')$ that

$$\{\mu'(y), \mu'(x)\} \in E(G) \iff \{v_j, a\} \in E(G) \iff \{w_j, b\} \in E(H) \iff \{\nu'(y), \nu'(x)\} \in E(H).$$

The middle equivalence follows from $atp(G, \bar{v}a) = atp(H, \bar{w}b)$. Similarly, it follows immediately from equivalence of the atomic types, that the vertex labels of $a$ and $b$ must be the same, and that $\kappa_G(\{v_j, a\}) = \kappa_H(\{w_j, b\})$. Additionally, we clearly have that $\mu', \nu'$ are in $(\bar{v}a, \bar{w}b)$-strict $S'$-agreement.

This concludes the argument for the individual homomorphisms. To see that this is indeed always a one-to-one correspondence it is enough to observe that because $c_i^k(\bar{v}) = c_i^k(\bar{w})$, for every distinct choice of $a$ there must be an appropriate distinct choice of $b$ as above. $\qquad\square$

**Lemma 12.** *Let $G$ and $H$ be labeled graphs, let $F$ be a labeled graph with a nice TD $(T^*, B)$ of width $k$, let $\bar{a}$ be a tuple made up of $k$ vertices that match a bag of the $(T^*, B)$. Let $\bar{v} \in V(G)^k$, $\bar{w} \in V(H)^k$ with $c^k(\bar{v}) = c^k(\bar{w})$. Then $|\mathsf{Hom}(F, G)[\bar{a} \mapsto \bar{v}]| = |\mathsf{Hom}(F, H)[\bar{a} \mapsto \bar{w}]|$.*

*Proof.* Let $r$ be a node in $T^*$ with $B(u) = \{a_i \mid i \in [k]\}$ and assume, w.l.o.g., that it is the root of $T^*$. We argue inductively on subtrees $T$ of $T^*$ that contain the node $r$ that there is a bijection $\iota$ from $\mathsf{Hom}(F[T], G)[\bar{a} \mapsto \bar{v}]$ to $\mathsf{Hom}(F[T], H)[\bar{a} \mapsto \bar{w}]$ such that for every $\mu$ in the domain of $\iota$, it and $\iota(\mu)$ are in colorful strict $T$ leaf agreement. Recall that colorful strict agreement requires agreement on colors (or color tuples) on some round $i \geq 1$ of the $k$WL algorithm. We will not explicitly argue these indexes in the following but instead note that because $c^k(\bar{v}) = c^k(\bar{w})$, which forms our base case in the following induction, we can start from assuming agreement at an arbitrarily high $i$. It will be clear from our induction that starting from an $i$ higher than the longest path from $r$ to a leaf will be sufficient.

In the base case $T$ contains only $r$. Since $c^k(\bar{v}) = c^k(\bar{w})$ we have that $\{v_i \mapsto w_i\}_{i \in [k]}$ is an isomorphism from $G[\bar{v}]$ to $H[\bar{w}]$. Thus, $\bar{a} \mapsto \bar{v} \in \mathsf{Hom}(F[T], G)$ iff $\bar{a} \mapsto \bar{w} \in \mathsf{Hom}(F[T], H)$.

For the step, assume the statement holds for a tree $T'$. Suppose we extend $T'$ be a node $\alpha$ that is adjacent in $T$, giving priority to split nodes (i.e., suppose we always extend first by all adjacent split nodes).

If $\alpha$ split node then the leaf bags remain unchanged. The only change that needs to be discussed is the special case that a new leaf is introduced. In that case, by definition the new leaf has the same bag as an existing leaf and the same witnesses for colorful strict $T'$ leaf agreement apply to the new bag as well, that is $\iota$ satisfies the desired property also for $T$.

If $\alpha$ is not a split node, then let $\beta$ be the leaf of $T'$ that is adjacent (in $T^*$) to $\alpha$. Suppose $\alpha$ is a forget node. By assumption any $\mu \in \mathsf{Hom}(F[T'], G)[\bar{a} \mapsto \bar{v}]$ and $\iota(\mu)$ are in colorful strict $T'$ leaf agreement. Since $B(T) = B(T')$, the sets of homomorphisms do not change $T'$ and $T$. It is therefore sufficient to check that the colorful strict leaf agreement constraints are also satisfied for leaf $B(\alpha)$. In that case, $\iota$ satisfies the desired properties also for $T$. Since $B(\alpha) \subseteq B(\beta)$, any $\mu, \nu$ that are in $(\bar{d}, \bar{e})$-strict $B(\beta)$-agreement, are also in $(\bar{d}, \bar{e})$-strict $B(\alpha)$-agreement. It remains to verify the color constraint on $\bar{d}, \bar{e}$. If $|B(\beta)| \leq k$, then we already have $c_i^k(\bar{d}) = c_i^k(\bar{e})$ by assumption and are done. Otherwise, we only know that $\mathsf{ct}(t, i, \bar{d}^\ominus) = \mathsf{ct}(u, i, \bar{e}^\ominus)$ for $\bar{d} = \bar{d}^\ominus t, \bar{e} = \bar{e}^\ominus u$. Let $z$ be the vertex from $B(\beta)$ that is forgotten by $\alpha$. Let $j$ be the index such that $\mu(z) = d_j$. We distinguish two cases: first, suppose that there is only one such index and there is at least one other $y \in B(\alpha)$ such that $\mu(y) = d_j$. Then there must be some index $j'$ such that $d_{j'}$ is not in $\mu(B(\alpha))$. By strict

agreement the same must hold for $e_{j'}$ and $\nu(B(\alpha))$. Hence, $\mu, \nu$ are also in $(\bar{e}[t/j'], \bar{d}[u/j'])$-strict $B(\alpha)$-agreement, and we have $c_{i-1}^k(\bar{e}[t/j']) = c_{i-1}^k(\bar{d}[u/j'])$ by equality of the color tuples. For the other case, if $z$ is the only vertex in $B(\alpha)$ that $\mu$ maps to $d_j$ or there is more than one index that equals $\mu(z)$, then simply take $j' = j$ and proceed as in the other case. This concludes the case where $\alpha$ is a forget node.

Suppose that $\alpha$ is an introduce node that introduces vertex $z$ and is adjacent (in $T^*$) to leaf $\beta$ of $T'$. By Proposition 10 we have that all vertices of $F[T]$ adjacent (in $F$) to $z$ are in $B(\alpha)$. By assumption there is a bijection $\iota'$ from $\mathsf{Hom}(F[T'], G)[\bar{a} \mapsto \bar{v}]$ to $\mathsf{Hom}(F[T'], H)[\bar{a} \mapsto \bar{w}]$ that maps every homomorphism to one that is in colorful strict $T'$ leaf agreement. Thus, in particular all homomorphisms are in $(\bar{d}, \bar{e})$-strict $B(\beta)$-agreement with $c_i^k(\bar{d}) = c_i^k(\bar{e})$ (because we assume width $k$, an introduce node can only follow nodes with at most $k$ vertices in the bag). We can therefore apply Lemma 11 to every pair $\mu, \iota(\mu)$ to see that there is a bijection $\iota'_\mu$ from $\mathsf{Hom}(F[T], G)[\mu]$ to $\mathsf{Hom}(F[T], H)[\iota(\mu)]$ such that every $\mu' \in \mathsf{Hom}(F[T], G)[\mu]$, is in colorful strict $T$ leaf agreement with $\iota'(\mu')$. If $|B(\alpha)| \le k$ we can apply the same reasoning as for forget nodes above to observe that the appropriate constraint on the color follows from the lemma.

Let $\iota'$ be the disjoint union of all the individual $\iota'_\mu$ for all $\mu \in \mathsf{Hom}(F[T'], G)[\bar{a} \mapsto \bar{v}]$. To see that the $\iota'_\mu$ are in fact disjoint it is enough to observe that the sets $\mathsf{Hom}(F[T], G)[\mu]$ are disjoint for different choices of $\mu$ as $dom(\mu) \subseteq V(F[T])$. Since all $\iota'_\mu$ are injective, so is $\iota'$. To see that $\iota'$ is also surjective, suppose a $\nu \in \mathsf{Hom}(F[T], H)[\bar{a} \mapsto \bar{w}]$ not in the image of $\iota'$. This means that $\nu$ cannot be in any set $\mathsf{Hom}(F[T], H)[\iota(\mu)]$ for $\mu \in \mathsf{Hom}(F[T'], G)[\bar{a} \mapsto \bar{v}]$. But $\iota$ is surjective and therefore this would mean that the projection of $\nu$ to $V(F[T'])$ is not a homomorphism from $F[T']$ to $H$ (extending $\bar{a} \mapsto \bar{w}$). But homomorphisms are closed under projection to induced subgraphs, so this is impossible. It follows that $\iota'$ is also surjective and therefore is the desired bijection. $\qquad\square$

As an alternative proof strategy, the main ideas from above can be used to define an algorithm that computes $\mathsf{Hom}(F, G)[\bar{a} \mapsto \bar{v}]$ from $c^k(\bar{v})$. An argument along this direction also demonstrate computability of the functions in Theorem 6 and Lemma 7 in the general case (in the uniform case where $\mathcal{C}^k$ is finite, it suffices to enumerate graphs until all colors in $\mathcal{C}^k$ are observed). However, in practice the colors in the $k$WL test are not maintained in their explicit form as nestings of tuples and multisets but rather by compactly representing the ct tuples via some hash function. In that setting, the viability of such an algorithm would be limited.

*Proof of Lemma 7.* Observe that for any two distinct tuples $\bar{v}, \bar{w} \in V(G)^k$ we have that the sets of homomorphisms $\mathsf{Hom}(F, G)[\bar{a} \mapsto \bar{v}]$ and $\mathsf{Hom}(F, G)[\bar{a} \mapsto \bar{w}]$ are disjoint. It is then straightforward to see that $\mathsf{Hom}(F, G) = \biguplus_{\bar{v} \in V(G)^k} \mathsf{Hom}(F, G)[\bar{a} \mapsto \bar{v}]$ and hence $\mathsf{homs}(F, G) = \sum_{\bar{v} \in V(G)^k} |\mathsf{Hom}(F, G)[\bar{a} \mapsto \bar{v}]|$. By Lemma 12 we have that $\mathsf{Hom}(F, G)[\bar{a} \mapsto \bar{v}]$ depends only on $c^k(\bar{v})$. This naturally induces a function $\eta_{F, \bar{a}}$ that map the possible colorings of the $k$WL test $\mathcal{C}$ to the respective number of homomorphisms, i.e., such that

$$\sum_{\bar{v} \in V(G)^k} \eta_{F, \bar{a}}(c^k(\bar{v})) = \sum_{\bar{v} \in V(G)^k} |\mathsf{Hom}(F, G)[\bar{a} \mapsto \bar{v}]| = \mathsf{homs}(F, G)$$

$\qquad\square$

Another proof detail of Lemma 12 that we would like to emphasize is that it is not strictly necessary for the stable color of $\bar{v}$ and $\bar{w}$ to be equal. Inspection of the proof reveals that it is sufficient for them to be equal for $c_i^k$, where $i$ is greater than the maximal number of introduce nodes on a path from $r$ to a leaf. Building on this observation, Lemma 12 actually also shows that especially for small patterns (the number of introduce nodes is of course no greater than the number of vertices) it is sufficient for the graphs to be equivalent under a limited number of steps of the $k$WL test, rather than equivalent in the stable color, to deduce that $\mathrm{hom}(F, \cdot)$ is the same in both graphs.

**Theorem 13.** *Let $F$, $G$, and $H$ be labeled graphs and let $n = |V(F)|$. Suppose that $G$ and $H$ have the same color multiset after $n$ steps of the $k$WL test. Then* $\mathsf{homs}(F, G) = \mathsf{homs}(F, H)$.

## B  Technical Details for Section 3

The goal of this section is to show that for any labeled graph $F$ that has treewidth greater than $k$, there are labeled graphs $G$ and $H$ such that $G \equiv_{kWL} H$ but $\mathsf{homs}(F, G) \neq \mathsf{homs}(F, H)$. Thus, $G$ and $H$ are witnesses to the fact that the $k$WL test cannot express the function $\mathsf{homs}(F, \cdot)$.

To do so, we show that two important results on the functions homs for unlabeled graphs still hold in the labeled setting. To distinguish more clearly between the labeled and unlabeled case we will use the term *plain graphs* to be those labeled graphs where the vertex- and edge-label alphabets are singletons. For technical reasons we will initially focus on *edge-labeled graphs*, which are labeled graphs where the vertex-label alphabet is a singleton. For labeled graph $G$, we write $\mathsf{plain}(G)$ for the plain graph obtained by making all edge and vertex labels of $G$ the same (without changing the vertices or edges of $G$).

The main property of homs that we are interested in is equivalence for fixed classes of graphs. Formally, let $\mathcal{F}$ be a class of labeled graphs. We write $G \equiv_{\mathcal{F}} H$ if for every $F \in \mathcal{F}$ it holds that $\mathsf{homs}(F, G) = \mathsf{homs}(F, H)$. Two families of graph classes will be particularly important in the following. We write $\mathcal{T}_k$ for the class of plain graphs with treewidth at most $k$, let $\mathcal{LT}_k$ be the class of labeled graphs of treewidth at most $k$, and let $\mathcal{ET}_k$ for the class of edge-labeled graphs with treewidth at most $k$.

The first result that we need is the labeled version of a classic result due to Dvořák (2010) and later rediscovered by Dell et al. (2018). We recall the lemma as stated in the main body.

**Lemma 5.** *For all labeled graphs $G, H$ we have $G \equiv_{kWL} H$ if and only if $G \equiv_{\mathcal{LT}_k} H$.*

We wish warn the reader about an unfortunate confluence of terminology at this point. Some of the literature that we build on also refers to labeled graphs. Unfortunately, our use and the uses in the immediately related literature are pairwise distinct and thus require extra care of the reader. We will add further clarification at points where this becomes directly relevant in this section.

For the second key result we need to recall what it means for a class of plain graphs to be homomorphism-distinguishing closed. A class of plain graphs $\mathcal{F}$ is *homomorphism distinguishing closed* if for every graph $F \notin \mathcal{F}$, there are graphs $G, H$ such that $G \equiv_{\mathcal{F}} H$ and $\mathsf{homs}(F, G) \neq \mathsf{homs}(F, H)$. The notion naturally extends to the setting of (edge-)labeled graphs by taking $\mathcal{F}$, as a class of (edge-)labeled graphs and every mention of $G$, and $H$ as (edge-)labeled graphs. To clearly distinguish the property in text going forward we will refer to the case where all involved graphs are labeled as *labeled homomorphism-distinguishing closed*, and analogously we use *edge-labeled homomorphism-distinguishing closed* when all graphs are edge-labeled. For plain graphs the following was very recently shown by Neuen Neuen (2023).

**Theorem 14** (Neuen (2023)). *The class $\mathcal{T}_k$ is homomorphism-distinguishing closed.*

Our second key ingredient will be to show the same holds in the labeled setting. That is, that $\mathcal{LT}_k$ is labeled homomorphism-distinguishing closed.

For our setting where every edge has exactly one label, we are not aware of any clear way to build directly on Theorem 14 to show the labeled versions. Our plan will therefore be to show that the machinery underlying the proof of Theorem 14 still works in our labeled setting.

The necessary machinery is due to Roberson (2022), who introduced the theory of oddomorphisms and showed their deep connection to homomorphism distinguishability. We recall the definition here for the labeled setting, note however that the definition does not respect labels, except implicitly through the definition of a homomorphism of labeled graphs. The neighborhood $N_G(v)$ of vertex $v$ in labeled graph $G$ is the set of adjacent vertices (ignoring any labels). Let $F$ and $G$ be labeled graphs and let $\phi$ be a homomorphism from $F$ to $G$. A vertex $a$ of $F$ is *odd/even* (w.r.t. $\phi$) if $|N_F(a) \cap \phi^{-1}(v)|$ is odd/even for *every* $v \in N_G(\phi(a))$. An *oddomorphism* from $F$ to $G$ is a homomorphism $\phi$ such that:

1. every vertex $a$ of $F$ is either odd or even with respect to $\phi$, and
2. for every $v \in V(G)$, $\phi^{-1}(v)$ contains an odd number of odd vertices.

A homomorphism $\phi$ is a *weak oddomorphism* if there is a subgraph $F'$ of $F$, such that $\phi|_{V(F')}$ is an oddomorphism from $F'$ to $G$.

For plain graphs, Roberson showed that any class of graphs that is closed under weak oddomorphisms, disjoint unions, and restriction to connected components is homomorphism-distinguishing closed. It is one main goal of this section to show that the same also holds for labeled graphs.

**Theorem 15** (Labeled Version of Theorem 6.2 (Roberson, 2022)). *Let $\mathcal{F}$ be a class of labeled graphs such that*

1. *if $F \in \mathcal{F}$ and there is a weak oddomorphism from $F$ to $G$, then $G \in \mathcal{F}$, and*

2. *$\mathcal{F}$ is closed under disjoint unions and restrictions to connected components.*

*Then $\mathcal{F}$ is labeled homomorphism-distinguishing closed.*

Before we move on to the proof of Theorem 15, we first observe that the class $\mathcal{LT}_k$ satisfies the conditions of the theorem. Indeed, Condition 15 is trivially satisfied as the treewidth of a graph is the maximum treewidth over the graph's connected components (and labels do not influence treewidth). For Condition 1 we can reuse the following key part of the proof of Theorem 14.

**Proposition 16** (Neuen (2023)). *Let $F$ be a plain graph in $\mathcal{T}_k$. If there is a weak oddomorphism from $F$ to $G$, then $G \in \mathcal{T}_k$.*

Recall that labels serve no explicit role in the definition of an oddomorphism beyond restricting the set of homomorphisms overall. Formalizing this observation we can show the same also for labeled graphs.

**Lemma 17.**

*Let $F \in \mathcal{LT}_k$ and $G$ be a labeled graph. If there is a weak oddomorphism from $F$ to $G$, then $G \in \mathcal{LT}_k$.*

*Proof.* First, observe that for labeled graphs $F, G$, if $\phi$ is an oddomorphism from $F$ to $G$, then $\phi$ is also an oddomorphism from $\mathsf{plain}(F)$ to $\mathsf{plain}(G)$. Now, if there is only a weak oddomorphism $\phi$ from $F$ to $G$, there is some subgraph $F'$ such that $\phi|_{V(F')}$ is an oddomorphism from $F'$ to $G$. Hence, it is also a oddomorphism from $\mathsf{plain}(F')$ to $\mathsf{plain}(G)$, meaning that where is a weak oddomorphism from $\mathsf{plain}(F)$ to $\mathsf{plain}(G)$.

We have that $tw(F) = tw(\mathsf{plain}(F)) \leq k$, and therefore by Proposition 16 $\mathsf{plain}(G) \in \mathcal{T}_k$. And therefore also $tw(G) = tw(\mathsf{plain}(G)) \leq k$, i.e., $G \in \mathcal{LT}_k$. $\square$

Assuming Theorem 15, we then immediately obtain the key technical result for homomorphism distinguishing closedness in labeled graphs.

**Theorem 18.** $\mathcal{LT}_k$ *is labeled homomorphism distinguishing closed.*

The rest of this section is organized as follows. We first verify that Theorem 15 holds in Appendix B.1. We then argue Lemma 5 in Appendix B.2. Finally, in Appendix B.3 we briefly note why Lemma 4 also holds in the labeled setting.

### B.1 PROOF OF THEOREM 15

We will closely follow the proof of Theorem 6.2 in (Roberson, 2022). We will see that adapting the initial definitions in the right way will actually leave most of the framework for plain graphs intact. In many situations the effect of the labels is on the argument is very subtle. For the sake of verifiability we therefore repeat the modified arguments for the critical path in the unlabeled setting here. We wish to emphasize that, if not stated otherwise, all proofs are effectively due to Roberson.

We first recall the construction of Roberson for the case of plain graphs, roughly following the presentation of Neuen (2023). Let $G$ be a plain graph and $U \subseteq V(G)$. Define $\delta_{v,U}$ as 1 if $v \in U$ and 0 otherwise. The graph $\mathsf{CFI}(G, U)$ is the graph with vertices

$$V(\mathsf{CFI}(G, U)) = \{(v, S) \mid v \in V(G),\ S \subseteq I(v),\ |S| \equiv \delta_{v,U} \bmod 2\}.$$

The graph has edge $\{(v, S), (u, T)\}$ in $E(\mathsf{CFI}(G, U))$ if and only if $\{v, u\} \in E(G)$ and $v, u \notin S\Delta T$.

The definition can be naturally adapted to the labeled setting by simply inheriting the labels of the original edges and vertices in $G$. To this end, let $G$ now be a labeled graph. We will define the labeled

graph $\mathsf{CCFI}(G, U)$ with $V(\mathsf{CCFI}(G, U)) = V(\mathsf{CFI}(G, U))$. For vertex $(v, S) \in V(G, U)$, we set the label $\lambda(v)$. For the edge-labeling function $\kappa$ edges we simply say that $\kappa(\{(v, S), (u, T)\}) = \kappa(\{v, u\})$. That is, $\kappa(\{(v, S), (u, T)\}) = c$ if and only if $\kappa(\{v, u\}) = c$ and $v, u \notin S \Delta T$.

We will show that this definition preserves the key properties of $\mathsf{CFI}$ graphs in the labeled setting. This might seem unintuitive as it ignores the labels on the sets $S \subseteq I(v)$ as well as the label of vertices in $U$. However, the key arguments are relative to homomorphisms on the graphs from which the $\mathsf{CCFI}$ construction is obtained. That is, these homomorphisms already follow the constraints imposed by the labelings. We will say that two vertices $u, v$ are *adjacent via label $i$* if there is an edge $\{u, v\}$ with label $i$.

**Lemma 19.** *Let $G$ be a connected labeled graph and let $U, U' \subseteq V(G)$. Then $\mathsf{CCFI}(G, U) \cong \mathsf{CCFI}(G, U')$ if $|U| \equiv |U'| \bmod 2$.*

*Proof.* Let $e = \{u, v\} \in E(G)$ and $U' = U \Delta e$. We show that $G_U \cong G'_U$. Let $\phi : V(\mathsf{CCFI}(G, U)) \to V(\mathsf{CCFI}(G, U'))$ be

$$\phi((a, S)) = \begin{cases} (a, S \Delta \{e\}) & a = u \text{ or } a = v \\ (a, S) & \text{otherwise} \end{cases}$$

As in the plain graph case, it is straightforward to verify that this is a bijection. Suppose that $(a, S)$ and $(b, T)$ are adjacent via label $i$ in $\mathsf{CCFI}(G, U)$, i.e., $\{a, b\} \in E_i(G)$ and $\{a, b\} \notin S \Delta T$. Let $S', T'$ such that $\phi((a, S)) = (a, S')$, $\phi((b, T)) = (b, T')$. If $\{a, b\} = e$, then $S' = S \Delta \{e\}$ and $T' = T \Delta \{e\}$, hence $S' \Delta T' = S \Delta T$. Thus, $\{a, b\}$ is also not in $S' \Delta T'$ and $\phi((a, S))$ is adjacent to $\phi((b, T))$. In the other case that $\{a, b\} \neq e$, then $S'$ and $T'$ will both, individually, contain $\{a, b\}$ exactly if $S$ and $T$, respectively, did. So again $\{a, b\} \notin S' \Delta T'$. In both cases, the labels of the edge in $\mathsf{CCFI}(G, U)$ and $\mathsf{CCFI}(G, U')$ is simply inherited from $\{a, b\}$. Thus, $\phi$ preserves labeled adjacency.

If $(a, S)$ and $(b, T)$ are not adjacent via label $i$ in $\mathsf{CCFI}(G, U)$, then either $\{a, b\} \notin E_i(G)$ or $\{a, b\} \in E_i(G) \cap (S \Delta T)$. In the first case, clearly $\phi((a, S))$ and $\phi((b, T))$ cannot be adjacent via label $i$ either. In the latter case, we have shown in our argument above that $\{a, b\} \in S' \Delta T'$ if and only if $\{a, b\} \in S \Delta T$. Since $\{a, b\} \in S \Delta T$, $\phi((a, S))$ and $\phi((b, T))$ will not be adjacent via any label in $\mathsf{CCFI}(G, U')$.

We have shown that $G_U \cong G'_U$ for $U' = U \Delta e$. This implies the lemma via the same final argument as in (Roberson, 2022, Lemma 3.2). □

As a consequence we can focus our investigation on two specific graphs $\mathsf{CCFI}(G) := \mathsf{CCFI}(G, \varnothing)$ and its *twist* $\mathsf{CCFI}^\times(G) := \mathsf{CCFI}(G, \{v\})$ for some arbitrary $v \in V(G)$.

As a next step we adapt (Roberson, 2022, Lemma 3.4) to the labeled setting. We first recall a definition from Roberson (2022) that carries over unchanged from the plain graph setting. For any $U \subseteq V(G)$, the mapping $(v, S) \mapsto v$ is a homomorphism from $\mathsf{CCFI}(G, U)$ to $G$. We will refer to this mapping as $\rho$ in the following. For a homomorphism $\phi \in \mathsf{Hom}(F, G)$ define

$$\mathsf{Hom}_\phi(F, \mathsf{CCFI}(G, U)) = \{\psi \in \mathsf{Hom}(F, \mathsf{CCFI}(G, U)) \mid \rho \circ \psi = \phi\}$$

and observe that the sets $\mathsf{Hom}_\phi(F, \mathsf{CCFI}(G, U)$ for $\phi \in \mathsf{Hom}(F, G)$ partition $\mathsf{Hom}(F, \mathsf{CCFI}(G, U))$.

The following statement is subtle in our context. Except for qualifying graphs $G$ and $F$ to be labeled, the statement is verbatim the same as Lemma 3.4 in (Roberson, 2022). That is, the labels are not explicitly considered in the system of equations. Note however that the equations are relative to some homomorphism from $F$ to $G$, which makes the second set of equations implicitly follow the labeling. The fact that this system of equations does not change is the key to requiring only few changes in the later lemmas, as the argument proceeds mainly algebraically via this system of equations.

**Lemma 20** (Labeled version of Lemma 3.4 (Roberson, 2022))**.** *Let $G$ be a connected labeled graph, let $U \subseteq V(G)$, and let $F$ be any labeled graph. For a given $\phi \in \mathsf{Hom}(F, G)$, define variables $x_e^a$ for all $a \in V(F)$ and $e \in I(\phi(a))$. Then the elements of $Hom_\phi(F, \mathsf{CCFI}(G, U))$ are in bijection with solutions of the following equations over $\mathbb{Z}_2$:*

$$\sum_{e \in I(\phi(a))} x_e^a = \delta_{\phi(a), U} \qquad \text{for all } a \in V(F) \tag{2}$$

$$x_e^a + x_e^b = 0 \qquad \text{for all } \{a, b\} \in E(F), \text{ where } e = \{\phi(a), \phi(b)\} \in E(G) \tag{3}$$

*Proof.* Suppose $x_e^a$ that are a solution of the system of equations in the statement of the lemma. For $a \in V(F)$, let $S(a) \subseteq E(G)$ be the edges incident to $\phi(a)$ (regardless of label) for which the variable $x_e^a = 1$, i.e., the set $\{e \in I(G) \mid x_e^a = 1\}$. Let $\psi$ be the mapping $a \mapsto (\phi(a), S(a))$.

We first show that $\psi \in \mathsf{Hom}_\phi(F, \mathsf{CCFI}(G, U))$. By Equation (2), $\psi(a)$ is guaranteed to be a vertex of $\mathsf{CCFI}(G, U)$ for all $a \in V(F)$. To see that $\psi$ preserves labeled adjacency, suppose $\{a, b\} \in E_i(F)$. Then $e = \{\phi(a), \phi(b)\} \in E_i(G)$ since $\phi \in \mathsf{Hom}(F, G)$. From Equation (3) it follows that $x_e^a = x_e^b$. Then, by construction of $\mathsf{CCFI}(G, U)$, $\psi(a)$ is adjacent via label $i$ to $\psi(b)$ exactly if $\{\phi(a), \phi(b)\} \notin S(a) \Delta S(b)$. Meaning that $\psi(a)$ is adjacent via label $i$ to $\psi(b)$ if $e$ is either in both $S(a)$ and $S(b)$ or in neither, which is true because $x_e^a = x_e^b$. Vertex labels are directly preserved via $\phi$.

The arguments for injectivity and surjectivity from the original proof (Roberson, 2022, Lemma 3.4) apply verbatim. $\qquad\square$

Since we can use the same system of equations (which does not explicitly consider labels) as used in Roberson (2022), the development from Section 3 of (Roberson, 2022) holds almost unchanged from here. For the sake of completeness we repeat the relevant parts for the proof of Theorem 15 here, including discussion on why the proofs are unaffected by the labels where relevant.

Let $G$ be a connected labeled graph, $F$ a labeled graph, and let $\phi \in \mathsf{Hom}(F, G)$. Let $R$ be the set of pairs $(a, e)$ such that $a \in V(F)$ and $e \in E(\phi(a))$. We define the matrices $A^\phi \in \mathbb{Z}_2^{V(F) \times R}$ and $B^\phi \in \mathbb{Z}_2^{E(F) \times R}$ as:

$$A^\phi_{b,(a,e)} = \begin{cases} 1 & \text{if } a = b \\ 0 & \text{otherwise} \end{cases} \tag{4}$$

$$B^\phi_{\{b,c\},(a,e)} = \begin{cases} 1 & \text{if } a \in \{b, c\} \text{ and } e = \{\phi(b), \phi(c)\} \\ 0 & \text{otherwise} \end{cases} \tag{5}$$

Let $\chi_U$ be the characteristic vector of $\phi^{-1}(U)$, i.e., the vector where the element corresponding to vertex $a \in V(F)$ is 1 if $a \in \phi^{-1}(U)$ and 0 otherwise (i.e., the vector $(\delta_{\phi(a),U})_{a \in V(F)}$. The system of equations from Lemma 20 can then be expressed as

$$\begin{pmatrix} A^\phi \\ B^\phi \end{pmatrix} x = \begin{pmatrix} \chi_U \\ 0 \end{pmatrix} \tag{6}$$

**Theorem 21** (Labeled version of Theorem 3.6 (Roberson, 2022))**.** *Let $G$ be a connected labeled graph, let $U \subseteq V(G)$, and let $\phi \in \mathsf{Hom}(F, G)$ for some labeled graph $F$. Then $\mathsf{homs}_\phi(F, \mathsf{CCFI}(G)) > 0$ and*

$$\mathsf{homs}_\phi(F, \mathsf{CCFI}(G, U)) = \begin{cases} \mathsf{homs}_\phi(F, \mathsf{CCFI}(G)) & \text{if Equation 6 has a solution} \\ 0 & \text{if Equation 6 has no solution} \end{cases}$$

*Proof Sketch.* The original proof (Roberson, 2022, Theorem 3.6) is a straightforward application of Lemma 20. The equations are exactly the same as in the unlabeled case and the argument works without modification in our setting. $\qquad\square$

**Theorem 22** (Labeled Version of Theorem 3.13 (Roberson, 2022))**.** *Let $G$ be a connected labeled graph and $F$ be a labeled graph. Then $\mathsf{homs}(F, \mathsf{CCFI}(G)) \geq \mathsf{homs}(F, \mathsf{CCFI}^\times(G))$. Furthermore, the inequality is strict if and only if there exists a weak oddomorphism from $F$ to $G$. If such a weak oddomorphism $\phi$ exists, there is a connected subgraph $F'$ of $F$ such that $\phi|_{V(F')}$ is an oddomorphism from $F'$ to $G$.*

*Proof Sketch.* The original proof of (Roberson, 2022, Theorem 3.13) works unchanged, although it might not be considered straightforward to observe that this is the case. We therefore repeat the key steps here with emphasis on the points where it needs to be observed that edge labels do not affect the argument. We wish to stress again that this proof is due to Roberson. We repeat it here to allow for a more targeted discussion of why labels have minimal effect on the argument, and to note necessary subtle differences.

If $\mathsf{homs}(F, \mathsf{CCFI}(G)) \neq \mathsf{homs}(F, \mathsf{CCFI}^{\times}(G))$, then by Theorem 21 there is a $\phi \in \mathsf{Hom}(F, G)$ such that Equation (6) over $\mathbb{Z}_2$ does not have a solution. The non-existence of such a solution is equivalent to the existence of solution to the system (the *Fredholm Alternative*)

$$\begin{pmatrix} (A^\phi)^T & (B^\phi)^T \\ (\chi_U)^T & 0 \end{pmatrix} \begin{pmatrix} z \\ y \end{pmatrix} = \begin{pmatrix} 0 \\ 1 \end{pmatrix} \tag{7}$$

where $U = \{\hat{u}\}$ contains the vertex such that $\mathsf{CCFI}^{\times}(G) \coloneqq \mathsf{CCFI}(G, \{\hat{u}\})$, $y$ is indexed by vertices of $F$, and $z$ is indexed by edges of $F$.

Since the argument creates a certificate for the non-equivalence of the homomorphism counts, it might be unintuitive that constructing a weak oddomorphism from this certificate still works in the presence of the additional constraints of the edge labels. However, the certificate is still relative to a $\phi \in \mathsf{Hom}(F, G)$ that respects the edge labeling and as we will see, this is sufficient for the argument to still hold.

Observe that a solution $(y, z)^T$ to this system satisfies $(A^\phi)^T y = (B^\phi)^T$. Let $O = \{a \in V(F) \mid y_a = 1\}$ and let $F'$ be the subgraph of $F$ with all vertices and edges $E(F') = \{e \in E(F) \mid z_e = 1\}$. The goal of the argument now is to show that $\phi$ is an oddomorphism from $F'$ to $G$ (and thus a weak oddomorphism w.r.t. $F$). Since $F'$ is a subgraph of $F$, it is clear that $\phi$ is still a homomorphism from $F'$ to $G$, regardless of labels. The arguments for the properties of the oddomorphism themselves are parity arguments on the number of adjacent edges between $\phi^{-1}(v)$ and $\phi^{-1}(u)$ for adjacent $v, u \in V(G)$. These arguments also hold unchanged in the labeled setting as the labeled homomorphism $\phi$ still preserves all adjacencies. Finally, $F'$ as constructed above may not be connected. As in the unlabeled case we can then replace $F'$ be a connected component by showing that if there is a weak oddomorphism $\phi|_{V(F')}$ for some subgraph $F'$, then we can assume w.l.o.g., that $F'$ is connected (Roberson, 2022, Lemma 3.12). $\qquad\square$

**Lemma 23.** *Let $F, G$ be labeled graphs. Then* $\mathsf{homs}(F, \mathsf{CCFI}(G)) > \mathsf{homs}(F, \mathsf{CCFI}^{\times}(G))$.

*Proof.* It is known that the identity id on $V(G)$ is an oddomorphism from $G$ to $G$ in the plain graph case (Roberson, 2022, Lemma3.14). The definition of oddomorphism does not take the labels of edges into account (only the neighborhoods of vertices, which are unaffected by labels), and id is still a homomorphism in the labeled setting. Hence, id must also be an oddomorphism from $G$ to $G$ in the labeled setting. The statement then follows immediately from Theorem 22. $\qquad\square$

The lemma also demonstrates that $\mathsf{CCFI}(G) \not\cong \mathsf{CCFI}^{\times}(G)$ for any $G$.

This concludes our replication of the key statements of (Roberson, 2022, Section 3). We will require one additional statement that is not necessary in Roberson's original argument for the proof of Theorem 15. For technical reasons we will require an labeled graph $G$ such that for some $F$ we can guarantee that $\mathsf{homs}(F, G) > 0$. For plain graphs this is straightforward by taking a large enough clique for $G$. In the labeled case we need an alternative gadget.

**Lemma 24.** *Let $\Sigma, \Delta$ be finite label alphabets and $n \geq 1$. There is a connected labeled graph $G$ such that for every labeled graph $F$ with at vertex/edge-label alphabets $\Sigma, \Delta$ and $n$ vertices,* $\mathsf{homs}(F, G) > 0$.

*Proof.* Let $\mathcal{C}$ be the set of all (up to isomorphism) labelings of the $n$ vertex clique with labels from $\Sigma$ and $\Delta$. Let $G^*$ be the disjoint union of all graphs in $\mathcal{C}$. To get the desired graph $G$, add a fresh vertex with arbitrary label to $G^*$ that is adjacent to an arbitrary vertex of each connected component (via an arbitrary edge label). Any $n$ vertex graph $F$ with labels from $\Sigma, \Delta$ is a subgraph of some element of $G' \in \mathcal{C}$, and hence $\mathsf{homs}(F, G') > 0$. Since $G'$ is a subgraph of $G$, also $\mathsf{homs}(F, G) > 0$. $\qquad\square$

We are now finally ready to prove Theorem 15. Again we can follow the original proof of Theorem 6.2 in (Roberson, 2022) very closely. We give the proof in full as some changes are made due to our streamlined presentation of the preceding statements.

*Proof of Theorem 15.* Let $G$ be a graph not in $\mathcal{F}$. Let $G_1, \ldots, G_\ell$ be the number of connected components in $G$ and let $J$ be the connected graph from Lemma 24 such that $\mathsf{homs}(G_i, J) > 0$ for

all $i \in [\ell]$. By Condition (2), there is at least one connected component of $G$ that is not in $\mathcal{F}$, thus assume w.l.o.g., that $G_1 \notin \mathcal{F}$.

Let $H$ be the disjoint union of $\mathsf{CCFI}(G_1)$ and $J$, and let $H'$ be the disjoint union of $\mathsf{CCFI}^\times(G_1)$ and $J$. We will show that $H \equiv_{\mathcal{F}} H'$ and $\mathsf{homs}(G, H) \neq \mathsf{homs}(G, H')$. For the former, suppose $F \in \mathcal{F}$ and connected. By Condition (1), there is no weak oddomorphism from $F$ to $G$, and thus $\mathsf{homs}(F, \mathsf{CCFI}(G_1)) = \mathsf{homs}(F, \mathsf{CCFI}^\times(G_1))$ according to Theorem 22. Because $F$ is connected each homomorphism into $H$ is either a homomorphism into $J$ or into $\mathsf{CCFI}(G_1)$, i.e., $\mathsf{homs}(F, H) = \mathsf{homs}(F, \mathsf{CCFI}(G_1)) + \mathsf{homs}(F, J)$. The same holds for $H'$, i.e., $\mathsf{homs}(F, H') = \mathsf{homs}(F, \mathsf{CCFI}^\times(G_1)) + \mathsf{homs}(F, J)$. Since both terms in both sums are equal we see that $\mathsf{homs}(F, H) = \mathsf{homs}(F, H')$. Since this holds for all connected $F \in \mathcal{F}$, by Condition (2), it holds for all $F \in \mathcal{F}$.

We move on to show that $\mathsf{homs}(G, H) \neq \mathsf{homs}(G, H')$. Fist, observe that $\mathsf{homs}(G, H) = \prod_{i=1}^{\ell} \mathsf{homs}(G_i, H)$ and similarly for $H'$. our goal will be to show that $\mathsf{homs}(G_i, H) \geq \mathsf{homs}(G_i, H') > 0$ for all $i \in [\ell]$, with at least one inequality being strict. For each $i \in [\ell]$ we have

$$\begin{aligned} \mathsf{homs}(G_i, H) &= \mathsf{homs}(G_i, \mathsf{CCFI}(G_1)) + \mathsf{homs}(G_i, J) \\ &\geq \mathsf{homs}(G_i, \mathsf{CCFI}^\times(G_1)) + \mathsf{homs}(G_i, J) = \mathsf{homs}(G_i, H') > 0 \end{aligned}$$

The equalities are by connectedness, as already argued above. The first inequality follows from Theorem 21. By Lemma 23 first inequality is strict for $i = 1$ and thus $\mathsf{homs}(G, H) > \mathsf{homs}(G, H')$. $\quad\square$

## B.2  LEMMA 5

In this section we will discuss why Lemma 5 holds in the labeled setting. From Lemma 7 we directly see that for labeled graphs $G, H$, $G \equiv_{kWL} H$ implies $\mathsf{homs}(F, G) = \mathsf{homs}(F, H)$ for every $F \in \mathcal{LT}_k$. We thus focus our attention on showing that $G \not\equiv_k H$ implies the existence of an $F' \in \mathcal{LT}_k$ such that $\mathsf{homs}(F', G) \neq \mathsf{homs}(F', H)$.

As a first step, we consider the logic $\mathcal{C}_{k+1}^L$ of first-order formulas with $k + 1$ variables and counting quantifiers[4] over the signature that contains unary relation symbols $U_\sigma$ for each label $\sigma$ in the vertex label alphabet $\Sigma$, and binary relation symbols $E_\delta$ for each label $\delta$ in the edge label alphabet $\Delta$. We add the superscript $L$ here to distinguish from the usual use of $\mathcal{C}_{k+1}$ for unlabeled graphs. Cai et al. (1992) famously showed that for graphs $G, H$ we have $G \vDash \varphi \iff H \vDash \varphi$ for all $\varphi \in \mathcal{C}_{k+1}$ if and only if $G \equiv_{kWL} H$. It is folklore that this also holds in the setting of labeled graphs. We in fact only need one direction here, namely if $G \not\equiv_{kWL} H$, then there is a $\varphi \in \mathcal{C}_{k+1}^L$ such that $G \vDash \varphi$ and $H \not\vDash \varphi$. It is straightforward to verify that the original argument for this direction is unaffected by labels (Cai et al., 1992, Theorem 5.4, case $\neg 1 \Rightarrow \neg 2$).

In the rest of this section we will show that the existence of such a formula $\varphi$ implies the existence of a labeled graph $F$ for which the homomorphism count into $G$ and $H$ differs. For this purpose we can adapt an argument by Dvořák (2010).

We first recall the key definitions from (Dvořák, 2010). A $k$-marked[5] labeled graph $G$ is a graph with a partial function $\mathsf{mark}_G : [k] \to V(G)$. The set of *active markings* $M_G$ is the subset of $[k]$ for which $\mathsf{mark}_G$ is defined. Suppose that $G, H$ are $k$-marked labeled graphs with $M_H \subseteq M_H$, we define $\mathsf{homs}(G, H)$ as the number of those homomorphisms $\phi$ that also preserve markings, i.e., where $\phi(\mathsf{mark}_G(i)) = \mathsf{mark}_H(i)$ for each active marking $i \in M_G$. A $k$-*marked labeled quantum graph* $G$ is a finite linear combination with real coefficients of $k$-marked labeled quantum graphs. We require all graphs in the linear combination to have the same set of active markings, which we refer to as $M_G$. The function $\mathsf{homs}$ is extended to the case where the first argument is a $k$-marked labeled

---

[4]For a detailed definition of first-order logic with counting quantifiers refer to Cai et al. (1992).

[5]What we call $k$-marked here is referred to as $k$-labeled in (Dvořák, 2010) and much of the literature. Unfortunately, this would clash with the other with our (also common) terminology for graphs with vertex and edge labels.

quantum graph $G = \sum_i^\ell \alpha_i G_i$ as $\mathsf{homs}(G, H) = \sum_i^\ell \alpha_i \mathsf{homs}(G_i, H)$. The treewidth of a quantum graph $\sum_i \alpha_i G_i$ is $\max\{tw(G_i) \mid \alpha_i \neq 0\}$[6].

A $k$-marked labeled quantum graph *models* a formula $\varphi$ for graphs of size $n$ if $M_G$ consists of the indices of the free variables of $\varphi$, and for each labeled graph $H$ on $n$ vertices with $M_G \subseteq M_H$:

- if $H \vDash \varphi$ then $\mathsf{homs}(G, H) = 1$, and

- if $H \nvDash \varphi$ then $\mathsf{homs}(G, H) = 0$.

From here on we follow the same strategy as Dvořák, but require some changes to correctly handle labeled graphs.

A *product* $G_1 G_2$ of two $k$-marked labeled graphs $G_1, G_2$ is constructed by taking the disjoint union of $G_1$ and $G_2$, identifying the vertices with the same marking, and suppressing parallel edges *with the same edge label*. That is, we allow the product to have self-loops and to have parallel edges with distinct labels. If there are vertices $v \in V(G_1), u \in V(G_2)$ with the same marking but different vertex labels (we say that $G_1, G_2$ are *incompatible*), the product is a single vertex with a self-loop (and arbitrary labels).

For quantum $k$-marked labeled quantum graphs $G_1 = \sum_i \alpha_{1,i} G_{1,i}$ and $G_2 = \sum_i \alpha_{2,i} G_{2,i}$, the product $G_1 G_2$ is defined as $\sum_{i,j} \beta_{i,j} G_{1,i} G_{2,j}$. The coefficient $\beta_{i,j}$ equals 0 is $G_{1,i} G_{2,j}$ is the empty graph, contains a loop or parallel edges with distinct labels, otherwise $\beta_{i,j} = \alpha 1, i \alpha_{2,j}$. The motivation for this distinction in the coefficients $\beta_{i,j}$ is that if $H$ is a labeled graph, then by definition it has no self-loops and any two vertices are connected only by one edge with one label, hence there can be no homomorphisms into $H$ from a product $G_{1,i} G_{2,j}$ that produces such situations. Note that in the situation where $G_{1,i}, G_{2,j}$ are incompatible at least one of $G_{1,i}, G_{2,j}$ must have 0 homomorphisms into $H$ which is why we (ab)use the self-loop as a failure state in this case. In particular, we have $\mathsf{homs}(G_1 G_2, H) = \mathsf{homs}(G_1, H)\mathsf{homs}(G_2, H)$ for every labeled graph $H$. We write $\mathbf{0}$ for a $k$-marked labeled quantum graph where all coefficients are zero.

**Lemma 25** (Labeled version of Lemma 6, (Dvořák, 2010)). *For each formula in $\varphi \in \mathcal{C}_{k+1}^L$, and for each positive integer $n$, there exists a labeled quantum graph $G$ of tree-width at most $k$ such that $G$ models $\varphi$ for labeled graphs of size $n$.*

*Proof.* We construct $G$ inductively on the structure of $\varphi$. The base cases significantly differ from those in the proof of (Dvořák, 2010, Lemma 6). However, once we have established the fact for the base cases, the inductive steps remain the same as they rely only on the property that $\mathsf{homs}(G_1 G_2, H) = \mathsf{homs}(G_1, H)\mathsf{homs}(G_2, H)$, as discussed above, and a technical lemma for quantum graphs (Dvořák, 2010, Lemma 5) that is not affected by labels.

If $\varphi =$ true, let $G$ be the empty graph, if $\varphi =$ false let $G = \mathbf{0}$. If $\varphi = U_\sigma(x_i)$, let $G$ be the graph with a single vertex $v$ with $\mathsf{mark}_G(i) = v$ and $\lambda(v) = \sigma$.

If $\varphi = x_i = x_j$, let $G = \sum_{\sigma \in \Sigma} G_\sigma$ (the sum ranges over the elements of the vertex label alphabet) where $G_\sigma$ is the $k$-marked labeled graph with a single vertex $v$, $\mathsf{mark}_G(i) = mathsf{mark}_G(j) = v$ and $\lambda(v) = \sigma$. For any specific $k$-marked $H$, the vertex marked as $i$ and $j$ will have only one vertex label. Hence, there will be a homomorphism from exactly one $G_\delta$ (where $\delta$ matches the label in $H$) and there are 0 homomorphisms for all other germs of $G$.

If $\varphi = E_\delta(x_i, x_j)$ there are two cases: if $i = j$, then let $G = \mathbf{0}$ as the predicate cannot be satisfied in a self-loop free $H$. If $i \neq j$, let $G = \sum_{\sigma, \sigma' \in \Sigma} G_{\sigma, \sigma'}$ where $G_{\sigma, \sigma'}$ is the graph with adjacent vertices $v$ and $u$, marked as $i, j$ and labeled with $\sigma, \sigma'$, respectively, and $\kappa(\{v, u\}) = \delta$. As above, for any given $H$, one term of $G$ will have 1 homomorphism into $H$, and the others will all have 0.

That is, for every base case we have shown that the appropriate $k$-marked labeled quantum graph exists. It is clear that in all the base cases the treewidth is 1. The rest of the induction proceeds exactly as in (Dvořák, 2010, Lemma 6). □

---

[6]Homomorphisms from quantum graphs can be seen as an alternative representation of graph motif parameters. Because of the two distinct natures of their uses in this paper and the disjoint connections to prior work we have decided to not unify the presentation of these two concepts here.

*Proof of Lemma 5.* As discussed above, we have that for labeled graphs $G, H$, $G \not\equiv_{kWL} H$ implies the existence of a $\phi \in \mathcal{C}_{k+1}^L$ such that $G \vDash \varphi$ and $H \not\vDash \varphi$. By Lemma 25, there is a $k$-marked labeled quantum graph $F$ with treewidth at most $k$ that models $\varphi$. Hence, $\sum_i \alpha_i \mathsf{homs}(F_i, G) = 1 \neq 0 = \sum_i \alpha_i \mathsf{homs}(F_i, H)$, which means there must be some $i$ such that $\mathsf{homs}(F_i, H) \neq \mathsf{homs}(F_i, H)$ (recall that the linear combination is always finite). Since $F$ has treewidth at most $k$, $F_i \in \mathcal{LT}_k$ and therefore $G \not\equiv_{\mathcal{LT}_k} H$. $\qquad\square$

### B.3 ON LEMMA 4 FOR LABELED GRAPHS

Seppelt (2023) originally showed Lemma 4 for unlabeled graphs. Here we briefly note why the lemma holds unchanged in the labeled case. In particular, the lemma relies on three facts.

1. First, that $\mathsf{homs}(F, G_1 \times G_2) = \mathsf{homs}(F, G_1)\mathsf{homs}(F, G_2)$. Where $\times$ is the *direct product* as defined in (Lovász, 1967).

2. Second, that the matrix $(\mathsf{homs}(K, L))_{K, L \in \mathcal{L}}$ is invertible, where $\mathcal{L}_n$ is the class of all graphs with at most $n$ vertices.

3. And finally, that if $G \equiv_{\mathcal{F}} H$, then also $G \times K \equiv_{\mathcal{F}} H \times K$ for all graphs $K$.

The first two points are classic results by Lovász (1967), which were originally shown for all relational structures. They thus hold unchanged also for labeled graphs. For the final point, recall that $G \equiv_{\mathcal{F}} H$ means that for every $F \in \mathcal{F}$ we have $\mathsf{homs}(F, G) = \mathsf{homs}(F, H)$. Then by the first point also $\mathsf{homs}(F, G \times K) = \mathsf{homs}(F, G)\mathsf{homs}(F, K) = \mathsf{homs}(F, H)\mathsf{homs}(F, K) = \mathsf{homs}(F, H \times K)$.

## C PROOF OF THEOREM 8

This section will make some slight departures from the terminology used in the main body of the paper. For a more focused presentation, we will first discuss the case for unlabeled graphs and afterwards show how to additionally handle labels afterwards. In the main body, the Spasm of a graph was defined as the set of all homomorphic images, but here it will be simpler to take an alternative (equivalent) perspective. A *quotient* $G/\tau$ of a graph $G$ is obtained by taking a partition $\tau$ of $V(G)$ and constructing the graph like $G$ but with all vertices in the same block of $\tau$ identified. That is, the vertices of $G$ are the blocks of $\tau$, and the incidence of a vertex in $G/\sigma$ is the incidence of all vertices in the corresponding block of the partition. It is a standard observation that the set of all quotients of $G$ without self-loops is precisely $\mathsf{Spasm}(G)$ (see, e.g., Curticapean et al. (2017)). It will be convenient to also consider the set of all quotients of $G$, i.e., including those with self-loops, which we will refer to as $\mathsf{Spasm}^\circ$.

The arguments of this section will revolve around formulas of *monadic second-order logic* (MSO). That is, formulas of first-order logic that additionally allow for quantification over unary predicates. For formal definition see, e.g., (Courcelle, 1990).

We will decide the treewidth of graphs by deciding whether they contain certain other graphs as minors. To that end we first define the notion of minors formally.

**Definition 1.** A *minor model* from $H$ into $G$, is a mapping $f : V(H) \to 2^{V(G)}$ such that:

1. $\forall u \neq v \in V(H)\ f(u) \cap f(v) = \varnothing$,

2. $\forall v \in V(H)\ G[f(v)]$ is connected, and

3. $\forall \{u, v\} \in E(H)$ there is an edge (in $G$) between some vertex in $f(u)$ and some vertex in $f(v)$.

We say that $H$ is a minor of $G$ is there a minor model from $H$ into $G$.

Checking whether graph $H$ is a minor of graph $G$ via an MSO formula is a standard construction. Interestingly, to check whether $H$ is a minor of *some quotient* of $G$ is simpler, at least in terms of formulating the question in MSO. The key insight here is that the second condition of minor models can effectively be ignored, since there will always be a quotient where all vertices of $G[f(v)]$ are

identified. Instead, it is enough to guarantee that the image is non-empty. In concrete terms, to check whether $H$ is a minor of a graph in $\mathsf{Spasm}^\circ(G)$ we will use the following MSO formula.

$$
\begin{aligned}
QuotMinor_H \equiv \quad & \exists X_1, \ldots, X_c \subseteq V(G) \; (\bigwedge_i |X_i| \geq 1 \; \wedge \bigwedge_{i \neq j} (X_i \cap X_j = \varnothing) \; \wedge \\
& \bigwedge_{\{v_i, v_j\} \in E(H)} \exists x \in X_i \; \exists y \in X_j ((x, y) \in E(G)))
\end{aligned}
$$

Intuitively, the sets $X_i$ correspond to the image of the minor model for vertex $v_i$ in $H$. The minor is of the quotient $\tau$ where $\tau$ extends the disjoint sets $X_1, \ldots, X_c$ by the singletons $\{v\}$ for all $v \in V(G)$ but not in any $X_i$ for $i \in [c]$.

**Lemma 26.** *Let $G$ and $H$ be graphs. Then $H$ is a minor of a quotient $G/\tau \in \mathsf{Spasm}^\circ$ if and only if $G \vDash QuotMinor_H$.*

*Proof.* Suppose $H$ is a minor of $G/\tau$ and let $h$ be the respective minor map from $H$ into $G/\tau$. Let $\mu$ be the edge surjective endomorphism (i.e., the homomorphic image) from $G$ into $G/\tau$ induced by the quotient (i.e., every vertex maps to the block in $\tau$ that contains the vertex). Let us refer to the vertices of $H$ as $v_1, \ldots, v_c$. The formula is satisfied when for every $i \in [c]$, the second-order variable $X_i$ is assigned to $\{w \in V(G) \mid \mu(w) \in f(v_i)\}$, that is, all vertices in $G$ that are mapped by $\mu$ to vertices in $f(v_i)$. Clearly, the sets $X_i$ are disjoint, as the images of $f$ are disjoint. Furthermore, we know by assumption that $f$ is a minor model that for every $\{v_i, v_j\} \in E(H)$, there is at least one edge $\{a, b\}$ with $a \in f(v_i)$, $b \in f(v_j)$. Since $\mu$ is edge surjective, there are $a', b' \in X_i$ such that $\{a', b'\} \in E(G)$ and $\mu(a') = a, \mu(b') = b$.

For the other direction now assume that the formula holds. Let $\tau_X$ be the partition of $V(G)$ induced by some satisfying choices of $X_1, \ldots, X_c$ (any vertex $v$ not in any of these sets corresponds to a singleton $\{v\}$ in the partition). Consider the graph $G/\tau_X$ and let us refer to the vertex corresponding to the block defined by $X_i$ as $u_i$. We claim that $f: v_i \mapsto \{u_i\}$ is a minor model of $H$ into $G/\tau_X$. Condition 1 and 2 of a minor model are trivially satisfied. For Condition 3, observe that if there is an edge $\{v_i, v_j\} \in E(H)$ but no edge $\{u_i, u_j\} \in E(G/\tau_X)$, then there can be no $x \in X_i, y \in X_j$ that have an edge between them in $E(G)$, hence contradicting the satisfaction of the final block of conjuncts in $QuotMinor_H$. $\qquad\square$

We can resolve the possibility of self-loops with a simple observation about minors of quotient graphs. A self-loop occurs in quotient $G/\tau$ if there is a block $B$ of $\tau$ that contains adjacent vertices $v, u$. W.l.o.g., assume for now that $v, u$ are the only adjacent vertices in $B$. If $H$ is a minor of $G/\tau$, it is also a minor of $G/\tau'$ where $\tau'$ is like $\tau$ but with $u$ as a singleton rather than in $B$. The vertex for $\{u\}$ in $G/\tau'$ will be adjacent to the vertex for block $B' = B \setminus \{u\}$. Contracting the vertices for $B'$ and $\{u\}$ will produce exactly $G/\tau$ without the self-loop at $B$. Iterating this idea shows that $G/\tau$ without self-loops is a minor of some loop-free quotient of $G$.

**Proposition 27.** *Let $G$ be a graph and let $H$ be a loop-free graph. Suppose $H$ is a minor of some graph in $\mathsf{Spasm}^\circ(G)$. Then $H$ is a minor of some graph in $\mathsf{Spasm}$.*

Robertson and Seymour famously showed that the graph minor relation is a well-quasi-ordering. Together with the classic observation that any well-quasi-ordering has a finite set of minimal elements this leads to the following standard result.

**Theorem 28** (Robertson & Seymour (2004)). *For every minor closed class of graphs $\mathcal{P}$, there exists a finite set of graphs $\mathcal{F}$ such that $G \in \mathcal{P}$ if and only if no $F \in \mathcal{F}$ is a minor of $G$.*

**Theorem 29.** *Deciding $\max\{tw(F) \mid F \in \mathsf{Spasm}(G)\} \geq k$ is feasible in fixed-parameter linear time when parameterised by $k$, and in linear time for fixed $k$.*

*Proof.* We first check whether $tw(G) \leq k - 1$. It is well known that this can be decided in linear time for fixed $k$ (Bodlaender, 1996). If not, then $tw(G) \geq k$ and the algorithm returns true.

The property of having treewidth at most $k - 1$ is closed under graph minors. By Theorem 28 there exists a *finite* set $\mathcal{F}$ of forbidden minors for class of graphs with treewidth at most $k - 1$. Furthermore, the set $\mathcal{F}$ depends only on the parameter $k$. Thus, we have that a graph $H$ has treewidth $\geq k$ iff it does

not have treewidth $\leq k - 1$ iff it has some graph of $\mathcal{F}$ as a minor. Combining with Proposition 27, we can therefore decide whether the treewidth in the spasm is at least $k$, by checking for every $F \in \mathcal{F}$ whether $F$ is a minor of any quotient of $G$. According to Lemma 26, this is equivalent to checking whether the formula

$$\phi \equiv \bigvee_{F \in \mathcal{F}} QuotMinor_F$$

is satisfied by $G$. By Courcelle's Theorem (Courcelle, 1990) we can decide $G \vDash \phi$ in time $f(k, \varphi) \cdot |G|$ where $k$ is the treewidth of $G$. Since $\varphi$ depends only on $k$ this is equivalent to $f(k) \cdot |G|$ and hence polynomial for fixed $k$. $\qquad \square$

As a final note we wish to point out that this result is primarily of theoretical interest. For most practical pattern sizes, naive enumeration of Spasm and checking the treewidth individually with state-of-the-art systems for treewidth computation is feasible. The bottleneck in practice is the size of the pattern, as enumerating the contents of Spasm via naive methods, e.g., enumerating all partitions, becomes challenging above 11 vertices (see Sequence A000110 in the On-Line Encyclopedia of Integer Sequences (OEIS Foundation Inc., 2023)).

**Adding Labels** We finally discuss the necessary changes for labeled graphs in the above argument, from which Theorem 8 then follows immediately. By definition, treewidth of labeled graphs is unaffected by the labels. As such we can simply ignore labels in checking for minors. Where the labels make a difference however, is in the set Spasm itself. Not all loop-free quotients are homomorphic images in the labeled case, but rather we require an additional restriction.

A quotient $G/\tau$ will not be a homomorphic image in labeled graphs in two cases. First, if a block of $\tau$ contains two vertices with different vertex labels, and second, if the quotient would create parallel edges with different labels. To lift the previous argument to the labeled case it is therefore enough to enforce these two extra conditions in the $QuotMinor_F$ formulas. To this end, recall that in a satisfying interpretation of the formula, the sets $X_i$ correspond precisely to the non-trivial blocks of the quotient for which $F$ is a minor. The two extra restrictions can then be enforced simply by conjunction with the following two formulas formula inside the scope of second order quantification (where $\Sigma, \Delta$ are the vertex- and edge-label alphabets, respectively).

$$\bigwedge_{i=1}^{c} \bigvee_{\sigma \in \Sigma} \forall x \in X_i \; U_\sigma(x) \tag{8}$$

$$\neg \bigvee_{\substack{i,j \in [c]^2 \\ i \neq j}} \exists x_1, x_2 \in X_i \; \exists y_1, y_2 \in X_j \left( \bigvee_{\substack{\delta, \delta' \in \Delta \\ \delta \neq \delta'}} (x_1, y_1) \in E_\delta(G) \wedge (x_2, y_2) \in E_{\delta'}(G) \right) \tag{9}$$

The term in 8 is easy to interpret, every block $X_i$ of the partition must have uniform vertex labels. The term 9 states that there are no two blocks $X_i, X_j$ such that there are edges with two different edge labels between the blocks, i.e., it is not the case that the quotient will have parallel edges with different labels. Relation $E$ in the original definition of $QuotMinor$ can trivially be replaced by the disjunction over all edge-label relations. Thus we can easily adapt our argument above to also decide the existence of minor models in Spasm of a labeled graph.

