# OpenReview forum: "On the Power of the Weisfeiler-Leman Test for Graph Motif Parameters"
_ICLR.cc/2024/Conference — ICLR 2024 poster_

### Official Review · Reviewer_Jgdr · 2023-10-30

**Soundness:** 3 good
**Presentation:** 3 good
**Contribution:** 3 good
**Rating:** 6
**Confidence:** 2

**Summary:**

The WL test for graph isomorphism is an iterative message-passing-like process on a graph, where in each round the nodes share structural information accumulated in previous rounds with their adjacent nodes. In recent years it has been studied in connection to the expressibility of graph neural networks. This paper studies the higher dimensional variant k-WL, where instead of nodes the information is passed between k-tuples of nodes. This variant has been associated with "higher-order GNNs", a generalization GNNs where correspondingly, messages are passed between sets of nodes. The paper is theoretical and proves some results about the "WL dimension" of graph substructures, which is the smallest k such the k-WL test can distinguish between graphs that have a different number of copies of that substructure. The results concern to variants like induced versus non-induced substructures, whether the number of copies can be computed, and whether the WL dimension of a substructures can be computed efficiently.

**Strengths:**

The WL test has gotten a lot of focus in the DL community due to its connections to GNN, and this line of literature is by now standard in the ICLR community, and this paper furthers this line of research by proving additional new results.

**Weaknesses:**

This seems generally like a combinatorics paper and it corresponds mostly with the combinatorics and not the ML literature. The connection between WL and GNNs is well-established by now, but the direction taken in this paper does not seem directly relevant to GNNs, but rather related mostly indirectly just due to generally being about WL. There is only fleeting reference to why any of the questions studied here bear on GNNs (and even then they concern the rather niche "higher-order" k-GNNs and not standard GNNs). Even though it is in scope for GNN-related venues and particularly ICLR, it is not all that clear that ICLR is where this paper would actually find its interested readership.

**Questions:**

N/A

---

> ### Author Response · Authors · 2023-11-15
>
> > Even though it is in scope for GNN-related venues and particularly ICLR, it is not all that clear that ICLR is where this paper would actually find its interested readership.
>
> We believe that our findings hold important implications for both the theoretical underpinnings and practical applications of GNNs, although we acknowledge that we may not have conveyed these implications explicitly enough. Primarily, owing to well-established connections between GNNs and the WL test, our results establish upper bounds on the capacity of GNNs to count subgraphs. Furthermore, as we have mentioned in our response to Reviewer 2nvm, we posit that our results can serve as valuable insights for the development of novel GNN architectures capable of capturing specific graph motif parameters. Presenting our results at ICLR can provide us with a valuable chance to engage in direct conversations about the theoretical and practical implications of our work with the people involved in designing GNN architectures. This primary motivation is the driving force behind our decision to submit our work to this conference.

---

> > ### Comment · Reviewer_Jgdr · 2023-11-22
> >
> > I thank the authors for their reply.

---

### Official Review · Reviewer_2nvm · 2023-10-31

**Soundness:** 4 excellent
**Presentation:** 3 good
**Contribution:** 2 fair
**Rating:** 6
**Confidence:** 4

**Summary:**

This paper studies the power of the k-Weisfeiler-Lehman (k-WL) algorithm for the natural and fundamental task of computing graph motif parameters (a weighted combination of subgraph counts). The motivation for studying the k-WL test is the connection to the expressive power of graph neural networks (GNNs).

Building on recent results for unlabeled graphs, this paper characterizes the class of graph motifs that can be computed by the k-WL test for labeled graphs (i.e., with edge and vertex labels). A central quantity is the WL-dimension of the graph motif parameter $\Gamma$, which is the smallest $k$ for which a k-WL test returns distinct outputs for graphs which induce distinct values of $\Gamma$. In Theorem 2, the authors show that the WL-dimension of $\Gamma$ is equal to the maximum treewidth of any of the subgraphs in the support of $\Gamma$. In Theorem 6, the authors show that the result of the WL test can be used to compute the value of $\Gamma$ as well (alone, the $k$-WL can only distinguish between different values of $\Gamma$). In their final set of results, the authors turn to the more specific question of subgraph counting. Theorem 8 shows that it can be determined whether a labeled graph has WL-dimension at most $k$ in polynomial time. Proposition 9 examines the popular task of counting graphlets, and shows that counting k-graphlets has WL-dimension k - 1.

**Strengths:**

This work extends recent results on the $k$-WL algorithm to the setting of labeled graphs. This is an important conceptual extension, as many graphs do have additional information that can be taken into account for learning tasks. Technical details are clear and seem correct.

The authors' results have clear implications for the ability of GNNs to compute subgraph counts. Together, Theorems 2, 6 and 8 show that one can determine an appropriate value of $k$ in polynomial time for the motif-counting task at hand, and can then compute it using a modification of the k-WL test. This is quite useful and may impact how one designs GNNs for these kinds of tasks.

**Weaknesses:**

- The novelty of this work isn't clear to me. While the extension to labeled graphs is well-motivated, the results and techniques seem to be obtained by simple / straightforward extensions of recent work for unlabeled graphs. Can you describe what the key roadblocks and innovations are in this setting? This would make it easier to judge the contributions of this work.
- No numerical experiments, but given that this is a theoretical paper, this is only a mild weakness.
- What are the implications for GNN design for various tasks, based on the results in your paper? These are only hinted at, but given the audience of this paper, it would be worthwhile to have some discussion about it.

**Questions:**

- Is $homs(F_i, G)$ defined somewhere?
- In the definition of tree decomposition on page 5, do you mean to write $\{u,v \} \subseteq \alpha(t)$?
- Is there any quick intuition for why treewidth is the right determinant of the WL-dimension?

---

> ### Author Response · Authors · 2023-11-15
>
> Due to the comment character limit we will adress the question regarding the key technical roadblocks in a separate comment.
>
> > What are the implications for GNN design for various tasks, based on the results in your paper? These are only hinted at, but given the audience of this paper, it would be worthwhile to have some discussion about it.
>
> Thank you for the suggestion. We agree that the paper would benefit from some further discussion of this point. We see the implications on two levels. High-level, our results resolve many questions around the theoretical limits of what can be expressed by GNNs.
> On a more concrete level, we believe that our results serve as a foundation to analyze, unify and shape the increasingly popular area of GNNs that are enriched with structural information about the input graph, such as Graph Substructure Networks (Bouritsas et al., 2022)  or GNNs with local graph parameters (Barceló et al., 2021).
> The motif parameter view can be seen as a list of pieces of information that are necessary to determine a function. It can thus allow us to precisely relate additional structural information that is injected into a GNN to the resulting change in expressivity.
> We will add respective discussions to the paper.
>
> > Is $hom(F_i, G)$ defined somehwere?
>
> Yes, in the last line of the penultimate sentence of page 3. We have defined Hom(F,G) for the set of homomorphisms from F to G and homs(F,G) for the number of such homomorphisms.
>
> > In the definition of tree decomposition on page 5, [...]
>
> Thanks! Indeed, there is a typo. It should be $u,v \in \alpha(t)$ or { $u,v$ } $\subseteq \alpha(t) $
>
> > Is there any quick intuition for why treewidth is the right determinant of the WL-dimension?
>
> High level, the key idea is that for any graph F with treewidth k, we know that if G and H are equivalent under kWL, homs(F,G) = homs(F,H) (due to Dvořák and Dell, Grohe, Rattan). At the same time the homomorphism distinguishing closedness property of treewidth implies that this is tight, in the sense that for any k’ < k there are graphs G,H that are k’WL equivalent but homs(F,G) \neq homs(F,H).
>
> For the first statement, the intuition is that kWL equivalence means (oversimplified) that the local view of at most (k+1) nodes at a time are the same in G and H. This means that in a tree decomposition for F, the possible local assignments of each bag for G and for H will be isomorphic. Since these local assignments in a decomposition determine the homomorphisms from F, the number will be the same.
>
> For the second statement we are unfortunately not aware of any convincing intuition. The argument is highly technical and relies on the combination of a number of elements that are already individually difficult to grasp intuitively (e.g., oddomorphisms, the algebraic characterisation of homomorphism mentioned above).

---

> > ### Comment · Reviewer_2nvm · 2023-11-22
> >
> > Thanks to the authors for their detailed responses.

---

> ### Author Response · Authors · 2023-11-15
>
> > The novelty of this work isn't clear to me. While the extension to labeled graphs is well-motivated, the results and techniques seem to be obtained by simple / straightforward extensions of recent work for unlabeled graphs. Can you describe what the key roadblocks and innovations are in this setting? This would make it easier to judge the contributions of this work.
>
> On a technical level the extension is not as straightforward as it seems in the high-level view. From Neuen’s proof of the unlabeled case we can reuse only the fact that the property of having treewidth at most k is closed under weak oddomorphisms. However, the underlying series of results by Roberson, from which it follows that this closure implies homomorphism distinguishing closedness needs to be extended to the labeled setting.
>
> This first requires us to adapt the construction of the CCFI graphs for the labeled case, which is non-trivial as there are multiple natural ways to generalize the unlabeled construction. In particular, in the unlabeled construction vertices are of the form (v,S) for all S that are  subsets of edges incident to v s.t. S satisfies some technical property. The generalization that we describe is arguably not even the most natural candidate as it ignores the edge labels in the definition of these vertices (the labels are of course used in a later, separate, part of the construction). However, it seems necessary to choose this less natural construction from a technical perspective.
>
> For this modified construction, we then need to reprove the key theorem of this oddomorphism framework. Notably, the main arguments are not combinatorial but instead rely on an algebraic characterization of homomorphisms into the CCFI constructions. The role of labels in this argumentation is thus less straightforward than one might expect and requires careful reproving (and subtle modifications) of a whole series of results. Furthermore, the construction of CCFI (and especially the algebraic characterisation) would be further complicated by the presence of vertex labels. We therefore technically split the argument in showing the case without vertex-labels, and then reducing the general labeled case to this case with only edge labels.
>
> In addition to this technical development which takes up a significant part of the technical appendix, Lemmas 4 and 5 also need to be lifted to their labeled version, which require different techniques (for Lemma 4 this is considered folklore in the context of parameterised counting, but the reasoning is not easily accessible to the intended audience of this paper). The arguments for Lemma 7 do not immediately build on any closely related results.
>
> Finally we believe that the Theorem 8 and Proposition 9 serve as illustrations for why we believe that this work serves as an important foundational contribution to the study of GNN expressiveness. Linking GNNs to recent work in counting complexity via the framework of graph motif parameters allows for relatively simple proofs of a notable open problem in the area (Thm 8), as well as an expressivity question that seemed entirely out of reach with previous methods (Prop 9).

---

### Official Review · Reviewer_p7nS · 2023-11-01

**Soundness:** 3 good
**Presentation:** 3 good
**Contribution:** 3 good
**Rating:** 8
**Confidence:** 2

**Summary:**

This work studies the Weisfeiler-Leman dimension of graph motif parameters for labeled graphs, i.e. the size of $k$ needed for $k$WL to distinguish these parameters. They show that this is exactly the maximum treewidth of the graphs in the support of the parameter. This solves several problems left open or unaddressed in recent work.

**Strengths:**

1. Uses results from, extends, and solves unsolved problems from very recent work on kWL and graph motif parameters.
2. Strong, elegant, general result on graph motif parameters, which allows application to many graph problems of interest, such as induced subgraph counting.

Due to these nice and timely theoretical results, I recommend acceptance of the paper.

**Weaknesses:**

1. Hard to understand introduction of first-order logic concepts
2. Unclear utility of results in machine learning: see questions section for notes on Section 4.

**Questions:**

Perhaps results from permutation-invariant function representation can be used to give a simpler proof of Theorem 6? It is known that multiset functions $f$ can be sum-decomposed in certain situations. For instance, [Zaheer et al. 2017] show that continuous permutation invariant functions $f$ from $[0, 1]^n \to \mathbb{R}$ (which can be viewed as functions on multisets of $n$ elements) can be written as $f(X) = \rho( \sum_{i=1}^n \phi(X_i))$ for some continuous functions $\rho$ and $\phi$. This motivates the common form of "readout" functions in GNNs, i.e. permutation invariant functions mapping from $k$-tuple representations to a single representation for the graph. In fact, from previous work I would already expect to see a result like yours; it is unclear whether your result is useful for practical GNNs, since one would already build a readout function in this form.


Notes:
1. Typo: "graph motif parameter by $ind_H$" on page 5
2. Typo: "counting counting", "of of" on page 8

---

> ### Author Response · Authors · 2023-11-15
>
> >Perhaps results from permutation-invariant function representation can be used to give a simpler proof of Theorem 6? It is known that multiset functions can be sum-decomposed in certain situations.. This motivates the common form of "readout" functions in GNNs, i.e. permutation invariant functions mapping from-tuple representations to a single representation for the graph. In fact, from previous work I would already expect to see a result like yours; it is unclear whether your result is useful for practical GNNs, since one would already build a readout function in this form.
>
> This is an interesting idea, but we are not in the position to provide a definite answer regarding its pertinence at the moment. About previous work: The only result we are aware of that relates to the task of counting the number of appearances of a pattern inside a graph is one given by Chen et al in the paper “Can graph neural networks count substructures?”. It is shown in such a paper that a GNN that performs local relational pooling over neighborhoods of the given graph G of radius at most the radius of the pattern p can count the number of occurrences of p in G. Our result is slightly different, as it shows that if the support of p only contains graphs of treewidth k then the number of occurrences of p in a graph G can be obtained from the result of a standard pooling function that counts the number of tuples of each color in the result of the WL test over G.

---

> > ### Comment · Reviewer_p7nS · 2023-11-22
> >
> > We thank the author for their reply!

---

### Official Review · Reviewer_kH24 · 2023-11-01

**Soundness:** 2 fair
**Presentation:** 2 fair
**Contribution:** 2 fair
**Rating:** 6
**Confidence:** 3

**Summary:**

The connection between the WL algorithm and the expressive power of GNNs is well-established in GNN literature. The paper conducts a theoretical study of the power and limitations of k-WL algorithm in context of counting patterns in graphs, more generally graph motif parameters. The authors provide a precise characterization of which labeled graph motif parameters are determined by the k-WL type of a graph. Some of these results concern induced versions of subgraph counting. Finally, the authors give a polynomial time algorithm that given a pattern P finds the minimum k such that k-WL determines P-counts.

**Strengths:**

The paper is a neat collection of results on pattern counting, which should be useful to the GNN community. The paper is extremely well-written and the exposition is very structured.

**Weaknesses:**

I am not convinced so much with the novelty of results, especially in light of the recent results such as Neuen (2023). The contributions are kind of incremental in my opinion. It would have been useful to have some experimental work to accompany the paper: e.g. Is it harder to compute labeled patterns using GNNs, instead of unlabeled patterns? Or perhaps, the performance of models using random node initialization could be explained by the ability of GNNs to count labeled patterns?

**Questions:**

1) Comment: Typically in homomorphism literature, a "labeled graph" refers to a graph with certain vertices marked with labels 1,...,k. That is, a k-labeled graph comes along with a mapping $\ell:[k] \to V(G)$. This is different from the more general notion considered in the paper. It would be advisable to make this distinction somewhere in the paper for readability across communities.

---

> ### Author Response · Authors · 2023-11-15
>
> > I am not convinced so much with the novelty of results, especially in light of the recent results such as Neuen (2023). [...]
>
> Please see our reply to Reviewer 2nvm, for a discussion of some of the key technical differences regarding Neuen (2023) specifically.
>
> > Comment: Typically in homomorphism literature, a "labeled graph" refers to a graph with certain vertices marked with labels 1,...,k. [...]
>
> Thank you very much for the suggestion. We indeed already make this remark in the appendix since we actually also use the notion described in the comment (although renamed to k-marked to avoid a clash of terms). But we agree that the potential for confusion is high and that this should be clarified already in the main body of the paper.

---

> ### Comment · Reviewer_kH24 · 2023-11-19
>
> Thanks for the answer: I have upgraded my rating from Marginally Below (5) to Marginally Above (6).
>
> I hope the authors will address the novelty aspects (differences to existing results) in their introduction more clearly. Considering the lack of experimental work/applications, I suggest that the authors should exactly pinpoint the relevance of their results to the GNN community in a separate paragraph within the introduction.

---

### Meta-Review · Area_Chair_gTHY · 2023-12-06

**Metareview:**

This paper studies how WL tests can encode graph motif parameters. Building on several recent theoretical work in graph theory, they extend these results from unlabeled graphs to labeled ones. This paper is clearly written and all reviewers are positive on it. Seeing this, I recommend acceptance following the majority.

**Justification For Why Not Higher Score:**

Most of the reviewers pointed out that this paper has limited novelty. Extending prior work from unlabeled graphs to graphs with discrete labels does not seem to be novel enough. Moreover, this paper may have limited accessibility due to the pure theoretical nature (also reflected by the confidence given by most reviewers).

**Justification For Why Not Lower Score:**

All reviewers gave positive scores.

---

### Decision · Program_Chairs · 2024-01-16

Accept (poster)